# Odor-identity dependent motor programs underlie behavioral responses to odors

Seung-Hye Jung[1], Catherine Hueston[1,2], Vikas Bhandawat[1,2,3]*

[1]Department of Biology, Duke University, Durham, United States; [2]Department of Neurobiology, Duke University, Durham, United States; [3]Duke Institute for Brain Sciences, Duke University, Durham, United States

**Abstract** All animals use olfactory information to perform tasks essential to their survival. Odors typically activate multiple olfactory receptor neuron (ORN) classes and are therefore represented by the patterns of active ORNs. How the patterns of active ORN classes are decoded to drive behavior is under intense investigation. In this study, using *Drosophila* as a model system, we investigate the logic by which odors modulate locomotion. We designed a novel behavioral arena in which we could examine a fly's locomotion under precisely controlled stimulus condition. In this arena, in response to similarly attractive odors, flies modulate their locomotion differently implying that odors have a more diverse effect on locomotion than was anticipated. Three features underlie odor-guided locomotion: First, in response to odors, flies modulate a surprisingly large number of motor parameters. Second, similarly attractive odors elicit changes in different motor programs. Third, different ORN classes modulate different subset of motor parameters.

*For correspondence: vb37@ duke.edu

**Competing interests:** The authors declare that no competing interests exist.

## Introduction

Humans rely chiefly on vision for their description of the world around them. But for many organisms, the world is dominated by their sense of smell, in which every day activities like finding food depends on the ability to modulate behavior based on olfactory cues. To meet the challenges of detecting and discriminating between different olfactory cues, animals are endowed with large families of odorant receptors (ORs) (*Spehr and Munger, 2009*; *Touhara and Vosshall, 2009*) that are expressed in olfactory receptor neurons (ORNs) and bind to odors. In phylogenetically diverse species, individual ORNs express only one or few ORs (*Buck and Axel, 1991*; *Su et al., 2009*) which largely determine their response specificity; thus ORNs can be classified into a discrete number of classes (*Vosshall et al., 1999*; *Vosshall et al., 2000*; *Clyne et al., 1999*; *Clyne et al., 1999*) according to the OR gene they express. Each odor activates multiple ORN classes and is represented by an ensemble of active ORN classes (*Hallem and Carlson, 2006*; *Hallem et al., 2004*; *Malnic et al., 1999*). How activities from different ORN classes are combined to modulate behavior is under intense investigation.

The clearest insights into odor modulation of behavior have come from the study of chemical communication. Initial work (*Karlson and Lüscher, 1959*) suggested that there are 'specialist' ORNs that bind to odors of particular ecological importance and play a major role in inter- and intra-species chemical communication. Recent work has shown that only a minority of chemical communication occurs through the detection of rare, highly-specialized chemicals by a single ORN class (*Kaissling, 1996*; *Dorries et al., 1995*). Most chemical communication involves integration of signals from multiple ORN classes (*Christensen and Sorensen, 1996*). Activation of different combinations of these ORNs can signal predator (*Mullersc, 1971*; *Endres and Fendt, 2009*), encourage approach (*Dorries et al., 1995*; *Dorries et al., 1997*; *Lin et al., 2005*; ) or aggression (*Ropartz, 1968*), cause

**eLife digest** Humans rely chiefly on vision to understand and navigate the world around them. But for many organisms, the world is dominated by their sense of smell. For these animals, everyday activities, like finding food, depend on being able to change behavior based on odor-based cues. To meet the challenges of detecting and discriminating between different odors, animals have many odorant receptors that bind to the odors, which are found on olfactory receptor neurons (ORNs). Each odor activates multiple ORNs, and different odors activate different combinations of ORNs. But it is not clear how activities from different classes of ORN are combined to create the perception of an odor or to guide behavior.

Now, Jung et al. have investigated the logic by which odors can alter a fruit fly's movements. The olfactory system of the fruit fly is organized along similar lines to that of a mammal, but is much simpler. Moreover, many genetic tools are available in fruit flies to allow neuroscientists to activate and inactivate specific neurons and assess the effect this has on behavior.

The results suggest that odor-guided movement in fruit flies has two noteworthy features. Firstly, in the presence of odors, flies alter their walking in unexpectedly large number of ways. Therefore, one needs to consider many different factors, or "motor parameters", to describe how odors affect a fly's movement. For instance, instead of just walking faster or slower, a fly can change how long it stops (stop duration), how long it runs (run duration) and how fast it runs (run speed) – all of which will affect overall speed. Secondly, a single class of ORN can strongly affect some parameters (like run duration) without affecting others (like stop duration). These data indicate that the neural circuits involved have a modular organization in which each ORN class affects a subset of motor parameters, and each motor parameter is affected by a subset of ORN classes. These findings were largely unexpected.

Jung et al.'s study focused on attractive odors. Future work will study repulsive odors to investigate if similar results are seen when studying repulsion versus attraction.

aggregation (*Bartelt, 1986*; *Bartelt, 1985*) and exert their effect by modulating a wide-range of specific motor programs implying that odors have diverse effect on an animal's behavior.

Most ORN classes are not involved in chemical communication but respond to a broad spectrum of chemicals and are 'generalists' (*Hildebrand, 1997*). According to current models of the function of generalist ORN classes, activity in the generalist ORNs are decoded by higher-order neurons to create an olfactory percept. Olfactory perception is probed by examining either an animal's ability to discriminate between different odors or the hedonic valence (*Knaden and Hansson, 2014*; *Knaden et al., 2012*) it assigns to different odors. These approaches to the study of olfactory perception are rooted in the psychophysical literature where an animal is made to choose between a few discrete behaviors (*Green, 1966*). A limitation of these approaches is that it has led to a 'portmanteau teleological' (*Kennedy, 1978*) description of generalist odors as attractants or repellents. As a result, even in a sophisticated analysis of an animal's navigation to an odor source, the experiments invariably test a single odorant (*Albrecht and Bargmann, 2011*; *Budick, 2006*; *Johnsen and Teeter, 1985*; *Weissburg, 1994*; *Baker and Kuenen, 1982*; *Porter et al., 2007*; *Gao et al., 2013*). The possibility that, like the ORNs involved in chemical communication, the generalist ORNs also signal diverse behavioral goals and exert their effect by modulating specific motor programs has not been explored.

In this study, using *Drosophila* as a model system, we directly assess whether generalist odors are classified into attractants or repellents or evoke a more diverse set of behaviors. We created a novel behavioral assay in which both the fly's level of attraction to an odor and the change in locomotion in the presence of that odor could be measured. We investigated how different odors which activate different ORNs modulate locomotion, and how mutating different ORN classes affects a fly's behavior in response to a natural odor. The null hypothesis was that based on the pattern of ORN activation flies would decide how attractive an odor is and modulate their locomotion according to the level of attraction. Our data is inconsistent with this simple model; and instead supports a different view of odor-guided locomotion that has three salient features. One salient feature is that odors independently modulate a surprising number of locomotor parameters. A second salient feature is

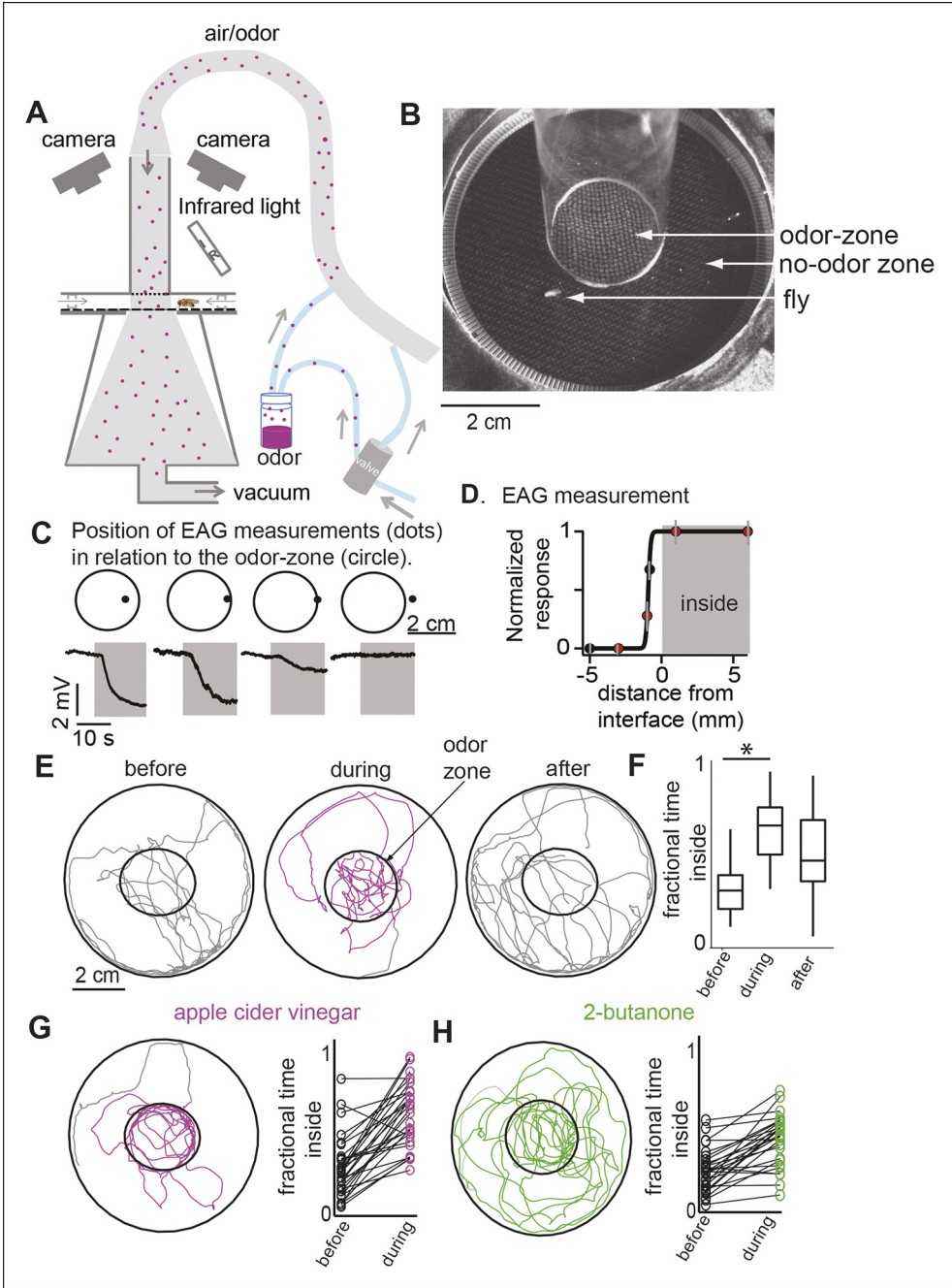

**Figure 1.** A novel behavioral paradigm for measuring odor-evoked change in fly's locomotion. (**A**) Schematic of the behavioral arena. (**B**) Top view of the chamber. (**C**) Electroantennogram (EAG) recording at different locations (indicated by a dot) shows a large EAG response when the measurement point lies within the odor-zone (denoted by circle). Response decreases when just the head of the fly is outside the odor-zone and is completely abolished 3 mm away from the odor-zone. (**D**) EAG response plotted as a function of distance from the nominal interface (n = 5). Red dots correspond to the data points shown in C. (**E**) Sample tracks of a fly in 3 min periods before, during and after presentation of ACV (left). (**F**) Boxplots (n = 29) showing that flies spend more time inside the odor-zone. (**G**) Track of another fly during ACV presentation. Connected dots represent the fractional time spent inside by a single fly before and during odor presentation (right). (**H**). Sample track of a fly during 2-butanone presentation shows that it is qualitatively different from the tracks in presence of ACV (left). Flies spend more time inside the odor-zone in the presence of 2-butanone (right, n = 31).

The following figure supplements are available for Figure 1:

*Figure 1. Continued*

**Figure supplement 1.** A detailed schematic showing the parts of the behavioral chamber.
**Figure supplement 2.** Flow visualization shows a precise interface between odor zone and no odor zone.
**Figure supplement 3.** Control experiments.

that two similarly attractive odors can produce changes in completely different aspects of locomotion. A third salient feature is that a single ORN class can strongly affect some motor parameters (like run duration) without affecting other parameters (like stop duration or angular speed). These data support a modular organization in which each ORN class affects a subset of motor parameters, and each motor parameter is affected by a subset of ORN classes.

# Results

## Behavioral assay

We designed a circular arena (*Figure 1A,B*; details in *Figure 1—figure supplement 1* and Materials and methods) in which the flies are constrained to walk between two plexiglass plates. A push-pull arrangement whereby air is pushed into the arena via an inlet tube and pulled through the arena by vacuum creates a sharp interface between a central zone of constant odor concentration (i.e. the odor-zone) and a surrounding no-odor zone. To demonstrate that the odor is limited to the odor-zone, we performed smoke visualization, and found that smoke introduced through the inlet tube was confined to the odor-zone, implying that odor from the inlet tube should also be limited to the odor-zone (*Figure 1—figure supplement 2*). To directly assess the spread of odors in the arena, we performed field-potential recordings from flies' antennae (i.e. electroantennogram or EAG) at different locations in the arena. The EAG responses were uniformly large inside the odor-zone and rapidly decreased with distance outside (*Figure 1C*). We estimate that the odor concentration decreases to less than 10% of its peak 3 mm away from the boundary of the nominal odor-zone (*Figure 1D*). Thus, in our walking arena an odor-zone is separated from a no-odor-zone by a sharp interface.

We measured the effect of odors on a fly's behavior by comparing its locomotion during a 3 min period immediately before the odor was turned on (i.e. the before period or control period) and a 3 min period in the presence of the odor (i.e. during period). Before odor onset, the fly spends most of its time at the outer border with occasional forays to the center of the arena (*Figure 1E*). The fly encounters the odor only when it enters the odor-zone for the first time after the odor is turned on. In the presence of apple cider vinegar (ACV), a strong attractant, flies spend 2.5-fold more time inside the odor-zone (*Figure 1F*) than during the control period. The fractional time inside is a measure of attraction. In later experiments, we will use a metric, attraction index (see Materials and methods), based on the change in time spent inside the odor-zone to measure attraction. This is comparable to the measures of attraction used in previous research (*Semmelhack and Wang, 2009*). The attraction to the odor remains high throughout the 3 min period when the odor is on (*Figure 1—figure supplement 3*) and there is no noticeable difference in the response of ORNs to ACV (*Figure 1—figure supplement 3*). The solvent used to dilute the odor, itself, did not result in a change in behavior (*Figure 1—figure supplement 3*).

## Two attractive odors affect different motor parameters

Do different attractive odors modulate a fly's locomotion in similar or distinct ways? In an initial screen for attractive odors, using fractional time spent inside as the metric for quantifying attraction, we found that 2-butanone was strongly attractive to the fly (*Figure 1H*, statistical test for attraction shown in *Figure 4*). We compared how undiluted ACV (ACV0) and 2-butanone at $10^{-3}$ dilution (BUN3), both of which are strongly attractive to the fly, modulate locomotion. The trajectory of a fly in response to the two odors is noticeably different (compare *Figure 1G–H*, also see *Video 1,2*). In the presence of ACV0, flies preferentially walk along the interface between the odor-zone and no-

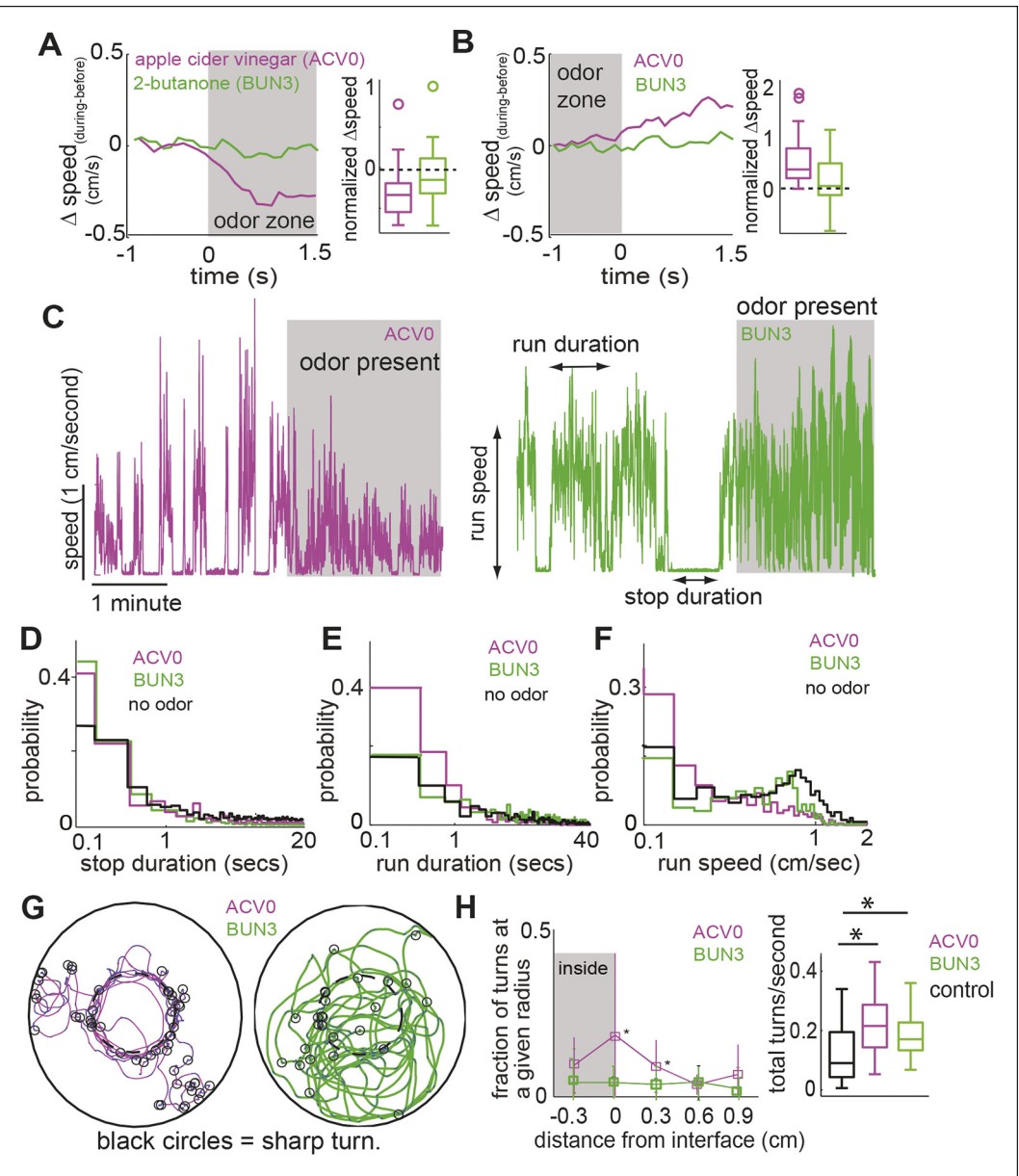

**Figure 2.** Different attractive odors modulate different motor parameters. (**A**) Flies decrease their speed when they enter the odor-zone in the presence of ACV0 but not BUN3. Left: Mean changes in speed between before and during periods. (n = 29 for apple cider vinegar (ACV0), n = 31 for 2-butanone [BUN3]). Right: Box plots showing the distribution of speed differences upon entering the odor zone. Normalization process in *Table 1*. (**B**) Same as in A except that change in speed as the fly leaves the odor-zone is plotted. (**C**) Sample trace showing speed as a function of time for a fly in response to ACV0 and another fly in response to BUN3. Stop duration decreases for both ACV0 and BUN3 but run duration and run speed only decreases for ACV0. Shaded regions represent the time during which odor is present . (**D,E,F**) Group statistics for stops and runs. (Black: before period (726 runs and 725 stops from 60 flies), magenta: ACV0 (n = 649), Green: BUN3 [n = 296]). Distribution of stop durations is significantly different for both ACV0 and BUN3 (p <10$^{-9}$ for both ACV and 2-butanone, KS test). Run duration and run speed is only different for ACV0 (p <10$^{-9}$ for ACV, p=0.08 for BUN3). (**G**) A fly preferentially executes sharp turns (marked by open black circles, see Materials and methods) at the interface between the odor-zone and no odor zone in the presence of ACV0 but not in the presence of BUN3. Traces are tracks of a single fly. (**H**) Left: Group data showing that flies preferentially execute turns right at the odor interface in ACV0 but not BUN3 (p <0.001 for both significantly different points). Right: Total turn frequency increases in response to both odors. Black: before period (n = 60), magenta: ACV (n = 29), Green: 2-butanone (n = 31).

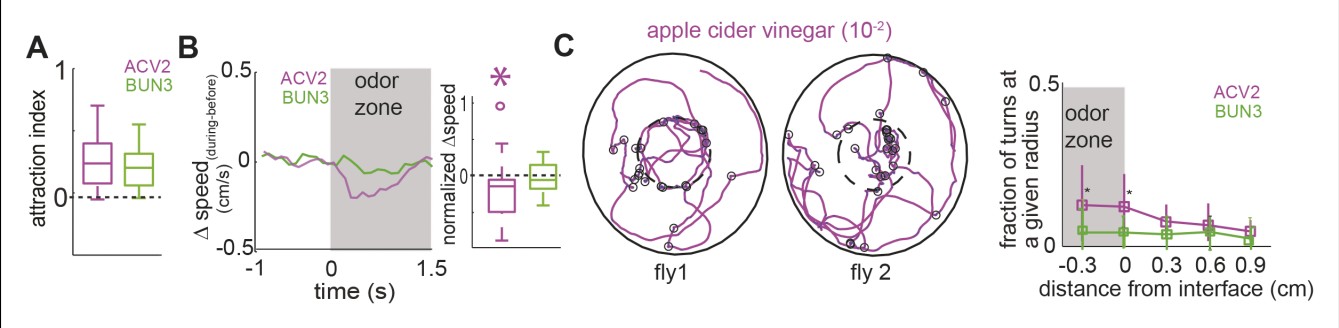

**Figure 3.** Two similarly attractive odors modulate different sets of motor parameters. (**A**) Attraction index showing that the fly is similarly attracted to two odors—apple cider vinegar ($10^{-2}$, ACV2, n = 34) and BUN3 (n = 31). Dotted line marks expected value when there is no odor modulation. (**B**) Decrease in speed upon entering the odor-zone is observed with ACV2 but not with BUN3 (left). Box plots showing the distribution of speed differences upon entering the odor zone (right). Normalization described in *Table 1*. (**C**) Left: 2 examples of walking trajectory in ACV2 trials during the presence of ACV2 with open black circles marking sharp turns. Right: Increased rate of sharp turns at the odor border is still observed with ACV2. 2-butanone data is replicated from *Figure 2* for comparison.

odor zone and make sharp turns outside the odor-zone to return to the odor-zone (*Figure 1E,G*). In contrast, in the presence of BUN3, the flies distribute more uniformly inside the odor-zone and return to the odor-zone largely via smooth turns (*Figure 1H*). Thus, visual observation of the differences in a fly's trajectory suggests that its response to BUN3 is not simply a scaled-down version of its response to ACV0. Rather, the differences in the fly's trajectories in response to the two odors suggest that different odors modulate a fly's locomotion in qualitatively distinct ways. One difference is the rapid change in speed as the fly enters the odor-zone. Flies decrease their speed when they enter the odor-zone in the presence of ACV0 but not in the presence of BUN3 (*Figure 2A* ; also see *Video 1*). Conversely, as the flies exit the odor-zone there is an increase in speed in the presence of ACV0 but not in the presence of BUN3 (*Figure 2B*).

Other differences in behavior are not immediately obvious and require further analysis (*Figure 2C–H*). One example is the odor-dependent modulation of how a fly partitions its time into bouts of walking and stopping (see Materials and methods (*Martin, 2004*; *Robie et al., 2010*). In the absence of odor, median stop duration is 0.67 s interspersed with occasional long ( >10 s) stops. In the presence of either odor, the long stops are largely abolished, and the median stop is only 0.3 s long (*Figure 2D*). In the absence of odor, median run duration is 4.81 s but runs frequently last >10 s (*Figure 2E*). However, the run duration (*Figure 2E*, median 1.0 s) and speed during runs (*Figure 2F*, median speed decreased from 0.57 cm/s to 0.18 cm/s) are only modulated in the presence of ACV0.

Modulation of turns by odor also differs between these two odors. In the presence of ACV0, flies make sharp turns (see Materials and methods for our definition) at the border between the odor and no-odor zone (*Figure 2G,H*). In the presence of BUN3, there is an overall increase in the frequency of turns in the presence of odor but the increase does not strongly peak at the odor interface.

One possible reason for the differences in behavior between ACV0 and BUN3 is that flies

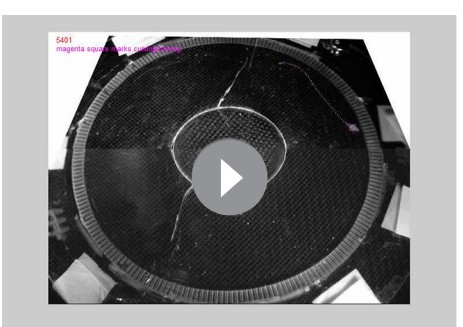

**Video 1.** This video shows the behavior of the fly to apple cider vinegar (*Video 1*). The tracks over the preceding 2 s are marked with dotted white line. The centroid is marked with green. The video also marks stops, sharp turns (S-turns) and curved walk. It shows that the fly slows down when it encounters apple cider vinegar (*Video 1*). Then, it explores the edge. This particular fly stays just outside the border for some time before entering. Other flies explore the border from the inside. Overall, the fly stays inside the odor-zone and explores the odor-zone with short frequent stops and increased turning.

**Table 1.** 17 behavioral parameter.
Parameters are defined in more details in the methods.

| Motor parameter | What it represents | How it is calculated (see Materials and methods for details) |
|---|---|---|
| Attraction index | Overall time spent inside the odor zone | $\frac{\text{(Time spent inside in the during period)}-\text{(Time spent inside in the before period)}}{\text{(Time spent inside in the before period)}}$ |
| Time spent/ transit | Median time spent inside the odor zone per visit | $\frac{\text{(median time inside per transit 'during')}-\text{(median time inside per transit 'before')}}{\text{(median time inside per transit 'before')}}$ |
| Time to return | Median time spent outside the odor zone between successive entries into the odor-zone | $\frac{\text{(median time outside per transit 'during')}-\text{(median time outside per transit 'before')}}{\text{(median time outside per transit 'before')}}$ |
| Radial density | Mean distance from the center of the arena | Fly's location as the distance from the center of the arena. Data were binned to 12 bins and normalized by area of each bin. |
| Speed inside | Mean speed inside the odor zone over the entire period | $\frac{\text{(average speed inside 'during')}-\text{(average speed inside 'before')}}{\text{(average speed inside 'before')}}$ |
| Speed outside | Mean speed outside the odor zone over the entire period | $\frac{\text{(average speed outside 'during')}-\text{(average speed outside 'before')}}{\text{(average speed outside 'before')}}$ |
| Speed crossing inside | Acute change in speed in the first 3 s after entering the odor-zone | $\frac{\text{speed crossing inside 'during'}-\text{speed crossing inside 'before'}}{\text{speed crossing inside 'before'}}$ |
| Speed crossing outside | Acute change in speed in the first 3 s after leaving the odor-zone | $\frac{\text{speed crossing outside 'during'}-\text{speed crossing outside 'before'}}{\text{speed crossing outside 'before'}}$ |
| Run duration | Average duration of runs | $\frac{\text{run duration 'during'}}{\text{run duration 'before'}}$ |
| Stop duration | Average duration of stops | $\frac{\text{stop duration 'during'}}{\text{stop duration 'before'}}$ |
| Run probability in | Fraction of time a fly spends running while inside the odor zone | $\frac{\text{probability that a fly is moving inside 'during'}}{\text{probability that a fly is moving inside 'before'}}$ |
| Run probability out | Fraction of time a fly spends running when outside the odor zone | $\frac{\text{probability that a fly is moving ourside 'during'}}{\text{probability that a fly is moving outside 'before'}}$ |
| Angular speed inside | Angular speed change inside the odor zone | $\frac{\text{(average angular speed inside 'during')}-\text{(average angular speed inside 'before')}}{\text{(average angular speed inside 'before')}}$ |
| Angular speed outside | Angular speed change outside the odor zone | $\frac{\text{(average angular speed outside 'during')}-\text{(average angular speed outside 'before')}}{\text{(average angular speed outside 'before')}}$ |
| Smooth turns in | Fraction of time a fly is performing a smooth turn inside the odor zone | $\frac{\text{Number of frames of smooth turns inside the odor zone}}{\text{Number of frames of run inside the odor zone}}$ |
| Smooth turns out | Fraction of time a fly is performing smooth turns outside the odor zone | $\frac{\text{Number of frames of smooth turns outside the odor zone}}{\text{Number of frames of run outside the odor zone}}$ |
| Sharp turns at boundary | Fraction of sharp turns near the odor boundary | The fraction of sharp turns which took place at the odor border, *i.e.* in a ring 2 mm around the border |

perceive ACV0 as more attractive than 2-butanone (median time spent inside the odor-zone in ACV0 is 1.5 times that in the presence of 2-butanone) and hence modulate more motor parameters in the presence of ACV0. If this rationale were true, we would expect that at a concentration at which the attractiveness of ACV0 and BUN3 are matched, the motor programs should be matched too. To test this idea, we performed behavioral experiments at 5 concentrations of apple cider vinegar and found that apple cider vinegar at a concentration of $10^{-2}$ (ACV2) had a similar level of attractiveness as BUN3 (*Figure 3A*). Yet, the motor parameters modulated by ACV2 and BUN3 are different. Unlike their response to BUN3, flies decreased their speed upon entering the odor-zone in the presence of ACV2 (*Figure 3B*). Moreover, the increased rate of sharp turns at the odor border observed with ACV0 was still observed at the lower concentration of ACV (*Figure 3C*). Thus, we concluded that equally attractive odors modulate distinct motor parameters.

Another line of evidence supports the idea that the level of attractiveness does not define the motor parameters modulated by a fly. Level of attraction, measured by the fraction of time a fly spends inside the odor-zone, depends on two mutually exclusive programs: Level of attraction would

increase if visits to the odor-zone last longer. This can be quantified by measuring time inside/transit (see Materials and methods). Level of attraction will also increase if the time between successive visits becomes shorter (time to return, see Materials and methods). If the level of attractiveness defines a fly's behavior, we would expect that flies that are more attracted to ACV0 should strongly modulate both time inside/transit ($t_i$) and time to return ($t_r$) and hence how strongly these two parameters are modulated in a given fly should be correlated. Instead, we found virtually no correlation between $t_i$ and $t_r$ (*Figure 4—figure supplement 1*), implying that there is considerable flexibility in the motor parameters modulated by a given odor. All together, the differences in motor parameters modulated by ACV2 and BUN3 and the lack of correlation between modulation of time inside/transit and time to return shows that level of attraction is a poor descriptor of the change in behavior in response to a given odor. In the rest of this study, we create a quantitative framework to analyze the behavioral response of the fly. Using this framework, we analyze the role of different ORN classes in mediating the behavioral response to ACV and the overall logic by which olfactory cues modulate locomotion.

## Odor-dependent changes in multiple motor programs

Olfactory behaviors have been characterized quantitatively in simpler organisms such as bacteria (*Berg and Brown, 1972*; *Pierce-Shimomura, 1999*); *C.elegans* (*Pierce-Shimomura, 1999*) and *Drosophila* larvae (*Gomez-Marin et al., 2011*). In all of these organisms, locomotion can be described in terms of straight runs punctuated by discrete stops during which the organism changes its orientation. Odors exert their effect on locomotion by modulating the duration of runs and by biasing orientation. This simple description fails to describe the behavior of an adult fly because a fly's locomotion cannot be decomposed into a series of straight runs and stops. For instance, flies can change their orientation by turning smoothly during a run. Our attempts to fit a statistical model such as Hidden Markov Model, which has been employed to describe the behavior of simpler organisms (*Gallagher et al., 2013*), were unsuccessful (data not shown). Therefore, we employed an ad-hoc approach to parameterize the fly's odor response and investigated how odors affect a large number of locomotor parameters Out of the parameters we investigated, we found 17 parameters (*Table 1*) that are all significantly (p <0.05 in a rank sum test) modulated by ACV0. These parameters describe how a fly's position (4 parameters), speed (4 parameters), run and stop statistics (4 parameters) and turns (5 parameters) are modulated (details in Materials and methods). We expected these parameters to be dependent on each other; and measured pairwise correlation to identify dependent parameters (*Figure 4—figure supplement 1*). Surprisingly, we found that the linear correlation between these 17 parameters is small, suggesting that they are largely independent of each other. The number of parameters required to describe a fly's behavior is large because a fly can modulate its locomotion inside the odor-zone independently from its locomotion outside it and also because the same parameter (like speed) is modulated independently across different time windows.

To estimate how these 17 parameters are modulated by ACV0 and BUN3, we performed a rank sum test with Bonferroni correction for multiple comparisons. The parameters which are significantly different at p <0.003 (or 0.05/17 - —corresponding to testing whether a given parameter is different after correcting for multiple comparison) were considered to be significantly modulated by a given odor (marked by a bar in individual panels in *Figure 4A*). Overlapping but distinct sets of parameters are modulated by ACV0 and BUN3 (*Figure 4A*). A fly's response to BUN3 is characterized by a decrease in stop duration and a large increase in its propensity to return to the odor-zone along gently curved trajectories (reflected as a change in smooth turns outside) which results in a large drop in $t_r$. The attraction of flies to BUN3 is

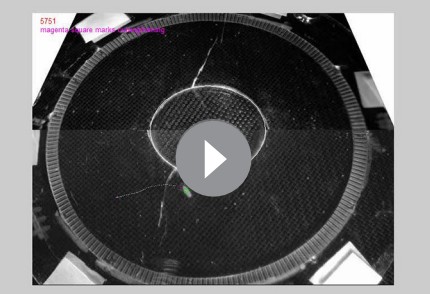

**Video 2.** This video shows the behavior of the fly to 2-butanone. The tracks over the preceding 2 s are marked with dotted white line. The centroid is marked with green. The video also marks stops, sharp turns (S-turns) and curved walk. Unlike in apple cider vinegar, although the fly keeps returning to the odor-zone, it often walks straight through it.

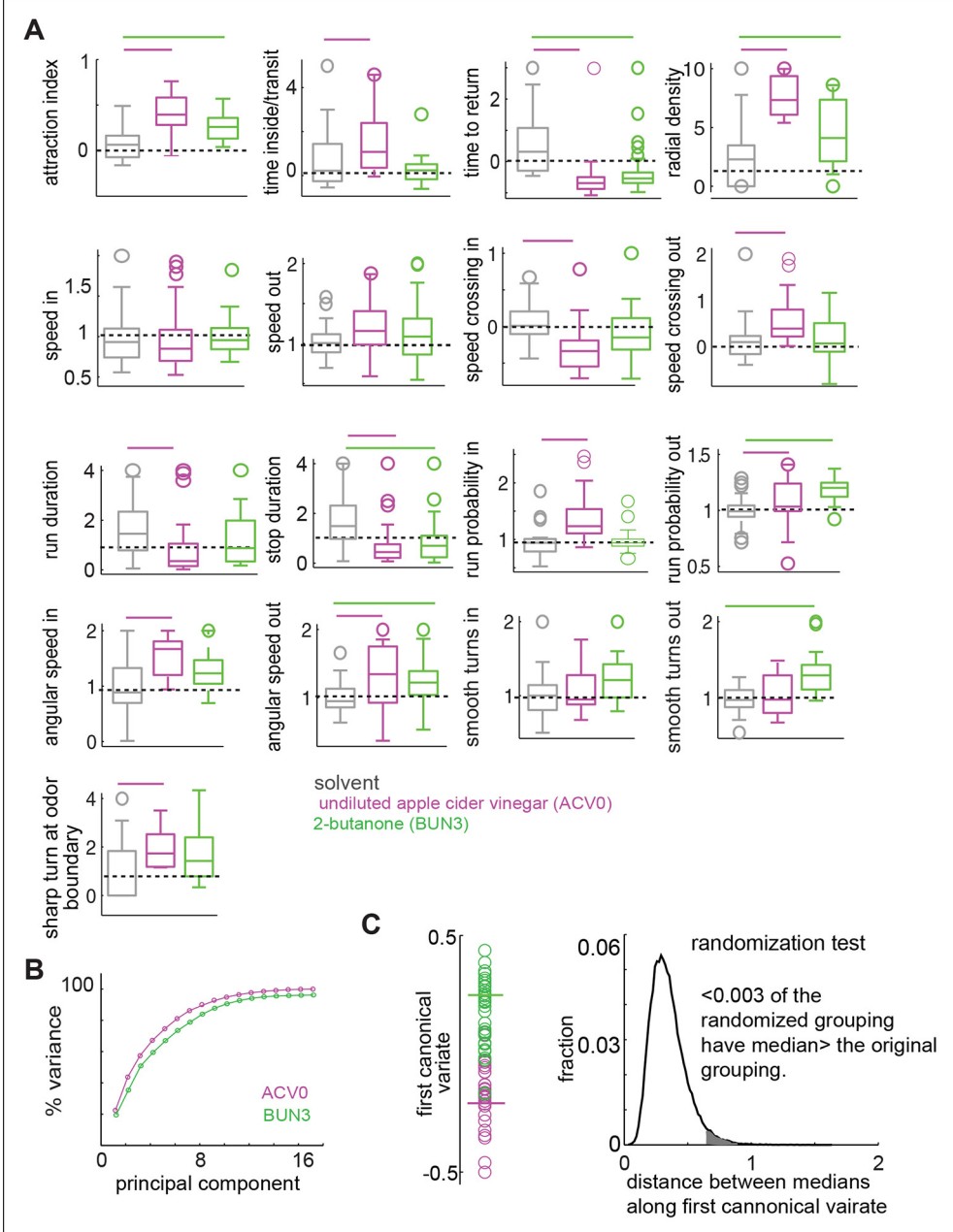

**Figure 4.** Odors independently modulate multiple behavioral parameters in an odor dependent manner. (**A**) 17 parameters which are all significantly modulated by ACV0 at p <0.05 (each parameter is described in detail in Materials and methods). Bars on top indicate the variables that are significantly different from the solvent control in a rank sum test after Bonferroni correction for multiple comparisons (p <0.003). Dashed line marks the expected value when there is no odor modulation. (**B**) Principal component analysis on the 17-dimensional odor space shows that ACV0 and BUN3 both activate multiple independent motor programs. (**C**) Left: The first canonical variates for the response of individual flies to ACV0 (magenta) and BUN3 (green). The responses due to BUN3 and ACV0 are clearly segregated along the first canonical variate. Each circle is a single fly. Right: The distribution of distances between medians in 50,000 trials in which odor labels were randomized. Less than 0.003 (0.3%) of the trials had medians greater than the original distribution (gray shaded area).

The following figure supplements are available for Figure 4:

**Figure supplement 1.** Evidence for independence of different motor parameters.

primarily a result of flies returning to the odor-zone more often. Flies decrease their return time even in the presence of ACV0, but they do so primarily by making sharp turns outside the odor-zone – an activity which leads them back to the odor. Additionally, the flies spend more time inside the odor-zone every time they enter inside by decreasing their speed inside the odor-zone and making sharp turns to stay inside it.

To test how many independent parameters are sufficient to describe a fly's behavior to odors, we performed principal component analysis (PCA) on the 17-parameter behavioral representation of an odor. PCA is a mathematical algorithm that reduces the dimensionality of data while retaining most of the variation in a particular data set. In an extreme case where the attractiveness of an odor determines how strongly every other parameter is modulated, we would expect that a single parameter (i.e. attraction index) would capture most of the variability in the behavior. Instead, we found that the first principal component only captured about 25% of the variance in data and the first seven principal components together contribute 90% to the variance in the data (*Figure 4B*). These results suggest that multiple independent parameters are necessary to describe a fly's behavior to an odor.

Using the representation of the behavior of a fly to an odor in the 17-dimensional behavioral space, we can demonstrate that the behavioral response to BUN3 is different from the behavioral response to ACV0. We performed the canonical variate analysis (CVA), which finds the axis in the 17-dimensional space which maximizes the ratio of between-group and within-group variances, i.e., finds the single dimension along which the behaviors due to the two odors is most different (see Materials and methods). Along the first canonical variate, the fly's response to ACV0 and BUN3 is clearly separable (*Figure 4C*). We performed a permutation test by randomly assigning responses to the ACV0 or BUN3 group and then performed CVA. The median distance between the randomly assigned ACV0 and BUN3 groups was larger than the median distance between the original groups in only 0.3% of the trials (*Figure 4D*).

We performed a similar analysis using the fly's response to ACV2 and BUN3. Consistent with the data presented in *Figure 3*, different parameters are modulated by ACV2 and BUN3 (*Figure 5*). Behavioral responses of a fly due to BUN3 and ACV2 are also separable along the first canonical variate, and support the hypothesis that two similarly attractive odors elicit very different motor responses (*Figure 5*). It is possible that the differences between ACV2 and BUN3 could be due to few flies which are strongly attracted to ACV. To control for this possibility we performed the same comparison, but with a subset of ACV2 flies whose distribution of attraction index closely matched that of BUN3. The differences in behavior between ACV2 and BUN3 still persist (*Figure 5—figure supplement 1*).

Taken together, the obvious visual differences in the trajectories of the fly in the presence of the two odors (*Figure 1F,G*), differences in modulation of individual parameters in different odors (*Figures 2,3*) and statistical analyses presented in *Figures 4,5* and *Figure 4—figure supplement 1* strongly support the idea that odors evoke changes in multiple motor parameters in an odor-dependent manner. It is possible that the differences in behavior arise from different temporal evolution of behavior. That is, the initial behavioral response is the same for all odors; but, behavioral response evolves differently for different odors resulting in behavior which is different for different odors. To control for this possibility, we evaluated the differences in behavior for BUN3 and ACV2 pair in the first 20 s after odor was turned on. We found that the differences in behavior between BUN3 and ACV2 are even greater in the first 20 s of exposure to odor (*Figure 5—figure supplement 1*). It is possible that the differences we observe between the two odors reflect the fact that ACV is a complex food odor and BUN is a monomolecular odor. To test for this possibility we compared the responses of the fly to ACV and banana. These responses are also distinct (*Figure 5—figure supplement 2*). Similarly, the response to BUN3 is different from the response to another monomolecular odor, ethyl acetate at $10^{-4}$ dilution (ETA4, *Figure 5—figure supplement 2*).

## Activation of a single ORN class results in modulation of a small subset of motor parameters

Like most other odors, ACV and BUN3 activate multiple ORN classes (*Semmelhack and Wang, 2009*; *Olsen et al., 2010*). To understand how activities from multiple ORN classes are integrated to drive behavioral response to natural odors, we carried out a series of experiments to dissect the contribution of different ORN classes to the behavioral response of a fly to ACV. We first assessed which

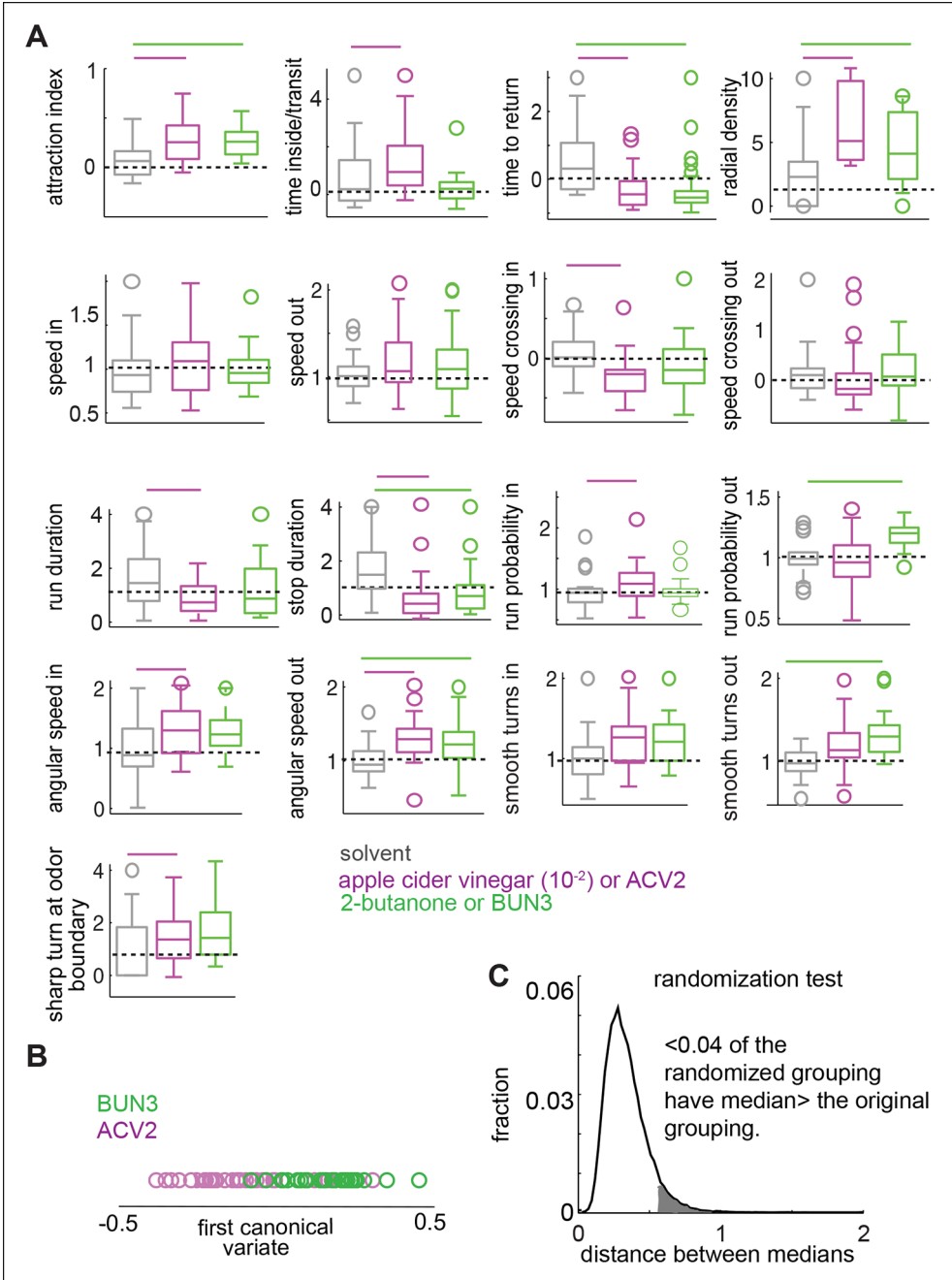

**Figure 5.** Two similarly attractive odors modulate different sets of motor parameters. (**A**) 17 motor parameters show that the parameters modulated by two similarly attractive odor—ACV2 and BUN3 are different. Bars on top indicate the variables that are significantly different from the solvent control in a rank sum test after Bonferroni correction for multiple comparisons (p <0.003). Dashed line marks the expected value when there is no odor modulation. (**B**) Canonical variate analysis shows that the behavioral response to ACV2 and BUN3 are distinct along the first canonical variate. (**C**) Permutation tests show that less than 4% of trials in which the odor labels are randomized had median distances greater than that of the original grouping.

The following figure supplements are available for Figure 5:

**Figure supplement 1.** Differences in behavior due to BUN3 and ACV2 is not due to a few flies which are very attractive to ACV or due to different temporal evolution of behavior in the two odors.

*Figure 5. continued on next page*

*Figure 5. Continued*

**Figure supplement 2.** Differences in the motor parameters modulated are not simply due to differences between simple and complex odors.

ORN classes are activated by ACV. ORNs are housed within three morphological classes of sensilla (*Shanbhag and Muller, 1999*)– basiconic, coeloconic and trichoid. Because most of the generalist ORN classes reside in the basiconic and coeloconic sensilla (*Su et al., 2009*), we measured odor-evoked responses from all basiconic and coeloconic sensilla using single sensillum recording. We matched the stimulus in the behavioral chamber to that on the electrophysiological rig by matching the amplitude of EAG response measured in the behavioral chamber to the EAG response in an electrophysiological rig. In all, we recorded responses from 10 basiconic sensilla which house 24 ORN classes, and 4 coeloconic sensilla which contain 11 ORN classes and from the antennal sacculus which houses Ir64a-ORNs. Out of these 36 ORN classes, we found that ACV0 activates 7 ORN classes (*Figure 6A* , see Materials and methods for details). Because we calculate most motor parameters as an average over the entire 3 min period, and because we have not observed any effect of different levels of activity of ORN on behavior, we are only reporting which ORNs are activated by ACV. In the rest of this study, we will use mutants, specific odors and lower concentration of ACV to probe the contribution of different ORN classes activated by ACV to behavior.

We first investigated how activation of a single ORN class modulates behavior. A previous study has shown that activating *Or42b*-ORNs alone produces robust attraction (*Semmelhack and Wang, 2009*). Moreover, we found that of the 7 ORNs activated by ACV, *Or42b*-ORNs are the most sensitive (see below). Therefore, we started by analyzing the role of *Or42b*-ORNs in behavior (*Figure 6B*). We activated the *Or42b*-ORNs alone by using low concentrations of ethyl acetate (*Olsen et al., 2010*). We have previously shown that ethyl acetate at a concentration $<10^{-6}$ only activates *Or42b*-ORNs[33,34] and confirmed that the same is true under the exact stimulus conditions used in this study (*Figure 6—figure supplement 1*). In behavioral experiments, we found that ethyl acetate at $10^{-8}$ (ETA8) does not elicit attraction (*Figure 6C*, first panel). The lack of attraction is not due to the activation of *Or85a*-ORNs (*Figure 6—figure supplement 1*) which are important for odor-mediated repulsion (*Semmelhack and Wang, 2009*). However, the lack of attraction does not imply a lack of behavioral response; we measured how ETA8 affects each of the 17 parameters and found that ETA8 elicits increase in angular speed and decrease in run duration compared to the solvent control (*Figure 6C*). Thus, activation of *Or42b*-ORNs alone affects only a small subset of motor parameters affected by ACV0, and modulation of these parameters does not lead to attraction. Moreover, both the increase in angular speed and decrease in run duration by ETA8 are abolished in the *Or42b* null mutant (*Figure 6—figure supplement 2*).

It is possible that the differences in behavioral response to ETA8 and ACV0 arise from the different firing rate these odors elicit in the *Or42b*-ORNs. This possibility is ruled out by the fact that ETA7 which elicits a similar firing rate as ACV2 (see *Figure 7*) elicits the same behavioral response as ETA8 (data not shown). In our behavioral paradigm, we did not observe any behavioral effect of increased firing rate in Or42b-ORNs on behavior.

We performed a similar experiment with another ORN class, *Or42a*-ORNs, which are also activated by ACV. To activate *Or42a*-ORNs alone, we used $10^{-5}$ dilution of 2-butanone (BUN5), a concentration at which it only strongly activates Or42a-ORNs (*Olsen et al., 2010*). We found that BUN5 also results in the modulation of a small number of motor parameters (*Figure 6D*). Importantly, BUN5 affects a distinct, but overlapping set of motor parameters compared to ETA8. Like ETA8, BUN5 affects angular speed inside the odor-zone. But unlike ETA8, it does not affect run duration; rather it causes a decrease in stop duration. No other parameters (of the 17 we investigated) other than the three shown in *Figure 6D* were affected when *Or42a*-ORNs were activated. These experiments suggest that each ORN class modulates a distinct but overlapping set of motor parameters.

## Attraction due to ACV results from a combination of different motor parameters modulated by different ORN classes

The modular organization in which different ORN classes activate different motor parameters makes two predictions: 1. the behavior of a fly at ACV concentrations which activates a single ORN class

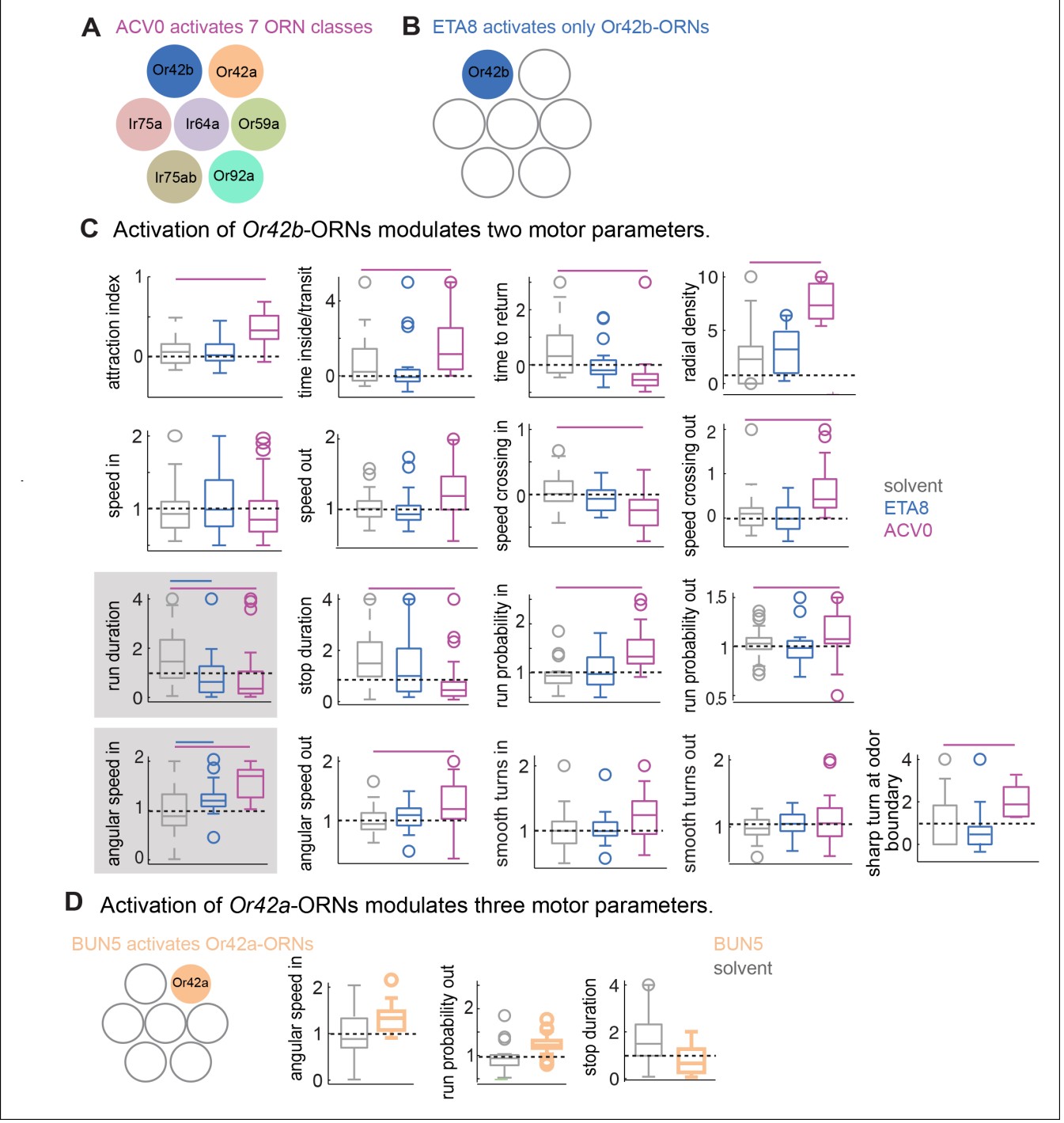

**Figure 6.** Activation of single ORN class leads to a change in a subset of motor parameters. (A) Schematic representation of ORNs activated by ACV0. (B) Ethyl acetate at low concentrations activate only Or42b-ORNs. (C) Behavioral modulation by activation of Or42b-ORNs alone using low concentration ($10^{-8}$) ethyl acetate (ETA8). ACV0 data is shown for comparison. Flies show increased angular speed inside the odor-zone and have a shorter run duration in response to ETA8 compared to the solvent control. Significantly modulated parameters (in a ranksum test after Bonferroni correction) are enclosed in a shaded box. (D) 2-butanone at $10^{-5}$ (BUN5) activates only Or42a-ORNs. Activating Or42a-ORNs results in modulation of three parameters. No other parameters were modulated. Only the significantly modulated parameters of the 17 are shown.

The following figure supplements are available for Figure 6:

**Figure supplement 1.** Ethyl acetate at low concentrations nearly saturates Or42b-ORNs without activating any other receptor.

*Figure 6. continued on next page*

*Figure 6. Continued*

**Figure supplement 2.** Behavioral response to ETA8 is abolished in Or42b mutants.

should be similar to the behavior when the same ORN class is activated by another odor, 2. Since higher odor concentrations activate more ORN classes we anticipate that as the odor concentration is decreased only a subset of motor parameters should be affected.

To test these predictions, we compared the behavioral response elicited by different ACV concentrations to the behavioral response elicited by activating the ORN class most sensitive to ACV. First, using the same strategy we employed to show that *Or42b*-ORNs are the only ones activated by low concentrations of ethyl acetate, we established that *Or42b*-ORNs are the most sensitive to ACV (data not shown). ACV elicits a robust response in *Or42b*-ORNs even at a $10^{-4}$ dilution (ACV4). EAG response to ACV4 is abolished in the *Or42b*-mutant implying that the *Or42b*-ORNs are the only ORNs strongly activated by ACV4. Next, in order to perform a direct comparison between the behavioral responses elicited by activating *Or42b*-ORNs alone to the behavioral response elicited by the entire complement of ORNs activated by ACV, we matched the spike rate elicited by ethyl acetate to that elicited by ACV (*Figure 7A*). ACV4 activates *Or42b*-ORNs alone and expectedly does not elicit attraction (*Figure 7B*). At concentrations below $10^{-6}$ (ETA7 and ETA8), ETA also activates *Or42b*-ORNs alone and fails to elicit attraction (*Figure 7B*). At higher concentrations ACV is attractive to the fly (*Figure 7B*) because it activates more ORN classes (see *Figure 7C–E*). ETA, too, at concentrations $10^{-6}$ or above activates multiple ORN classes. Consistent with the activation of more ORN classes, flies spend more time inside the odor-zone in the presence of ETA6.

As in the case of ETA8 (*Figure 6C*) lack of attraction does not imply a lack of behavioral response; ACV4 affects the same two parameters as ETA8 - run duration and angular speed inside (*Figure 7C*). ACV3 activates *Or59b*-ORNs in addition to *Or42b*-ORNs. Behaviorally, ACV3 modulates more motor parameters (*Figure 7D*); this is likely due to the activation of more ORN classes. Importantly, run duration and angular speed inside continue to be modulated because *Or42b*-ORNs are still activated. In addition, ACV3 also elicits a decrease in the time to return, a likely consequence of increased turn-rate outside the odor-zone. The number of parameters modulated increases further when the concentration is increased to ACV2. At this dilution, 10 parameters are affected (*Figure 7E*) and the behavior appears qualitatively similar to the fly's behavior to ACV0. These data further support the hypothesis that different ORN classes modulate different motor parameters. At low odor concentration when only *Or42b*-ORNs are activated by the odor, only run duration and angular speed in the odor-zone is affected. At higher odor concentrations, consistent with the activation of more ORN classes more motor parameters are modulated.

## Only a small subset of motor parameters modulated by ACV are affected in the *Or42b* mutants

Another prediction of our model is that if we perturb the activity in a subset of ORN classes, we should observe a change in a subset of motor parameters. To investigate this issue, we studied how the fly's ACV-evoked behavior changes in *Or42b*[EY14886] flies, a previously characterized null mutation in the Or42b receptor (*Bhandawat et al., 2007*). *Or42b*-mutants were as attracted to ACV as control flies (*Figure 8B,E* -attraction index panel, p >0.1 on a ranksum test); but some of the individual motor parameters were affected in the mutant. Two examples are shown in *Figure 8C,D*. The mutant flies had a significantly sharper distribution at the odor border (*Figure 8B, C*). The change in run duration was significantly diminished in the mutants compared to the control flies (*Figure 8D,E*). Based on the fact that activating Or42b-ORNs resulted in an increase in angular speed (*Figure 6C,7C*), we expected a decrease in angular speed inside in the *Or42b* mutants. Although the median angular speed inside does decrease in the mutant, this decrease in angular speed is not significant. The lack of significant effect on angular speed implies that multiple ORNs activated by ACV can modulate angular speed. Overall, only 3 parameters out of 17 we investigated were affected in the mutants (*Figure 8E*). To confirm that these differences are odor-dependent and do not simply represent differences between the two genotypes, we compared the responses of control and mutant flies to the solvent control and found no differences ( *Figure 8—figure supplement 1*). Thus, consistent with experiments in which we activated *Or42b*-ORNs (*Figure 6*), a null mutation in

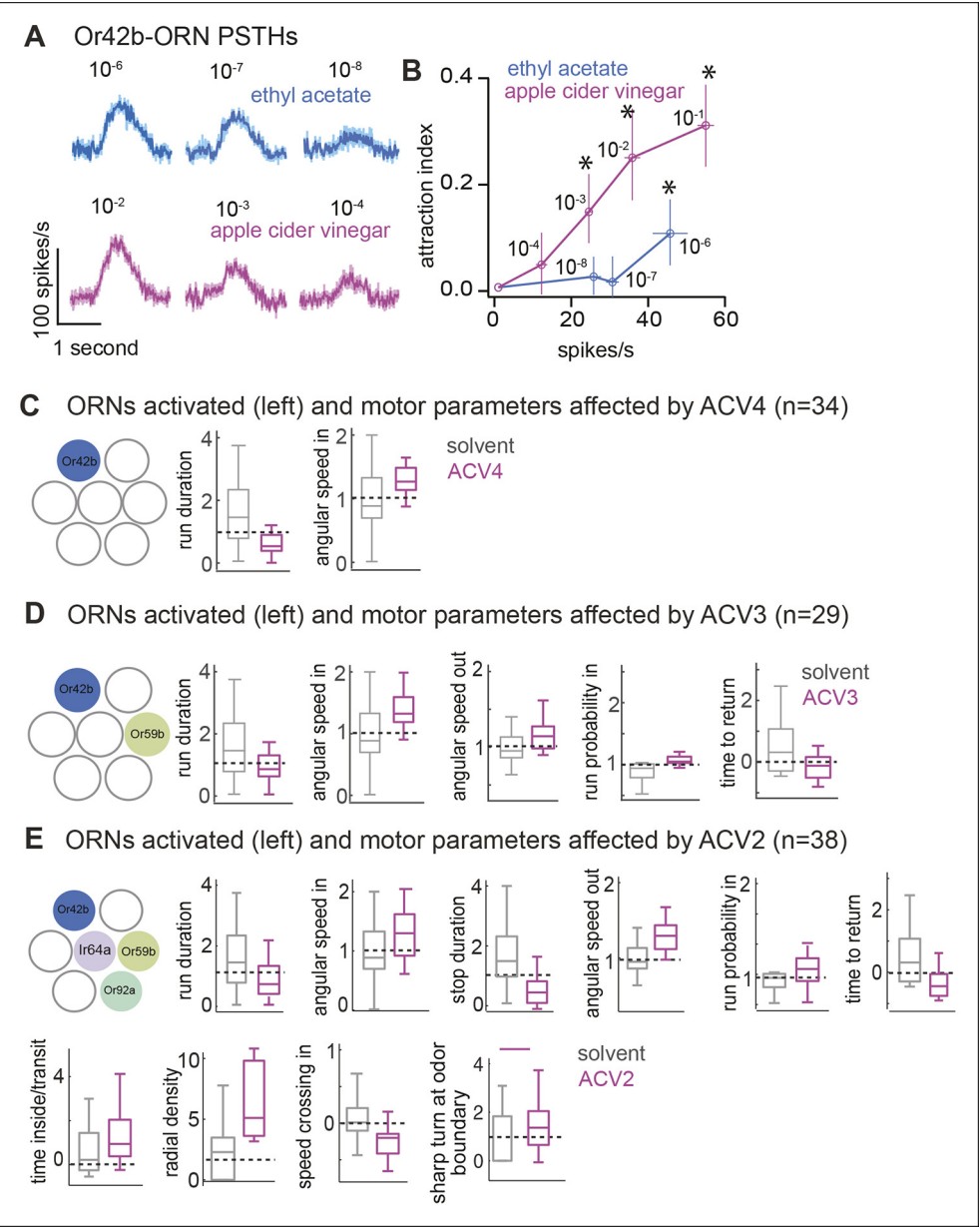

**Figure 7.** Higher concentration of ACV activates more ORNs and recruits more motor parameters. (**A**) PSTHs showing the response of the Or42b-ORN to ACV and ethyl acetate (mean ± SEM, n = 5-7). (**B**) At all spike rates, ACV is more attractive than ETA implying that attraction due to ACV is not due to activation of Or42b-ORNs alone. (**C–E**) Motor parameters modulated by increasing concentration of ACV. Only parameters that are modulated are shown.

the *Or42b* gene does not abolish the attraction of a fly to ACV. Instead, it selectively affects a small fraction of motor parameters.

## Modulation of behavior inside the odor-zone is strongly affected in the *Ir8a* mutants

Since different ORN classes affect distinct motor parameters, inactivating multiple ORN classes should affect the modulation of more parameters than a single ORN class. Out of the 7 ORN classes activated by ACV0, 3 ORNs require a co-receptor, Ir8a. In the *Ir8a* mutant (*Abuin et al., 2011*), these 3 ORN classes are non-functional and only 4 ORN classes activated by ACV0 are functional

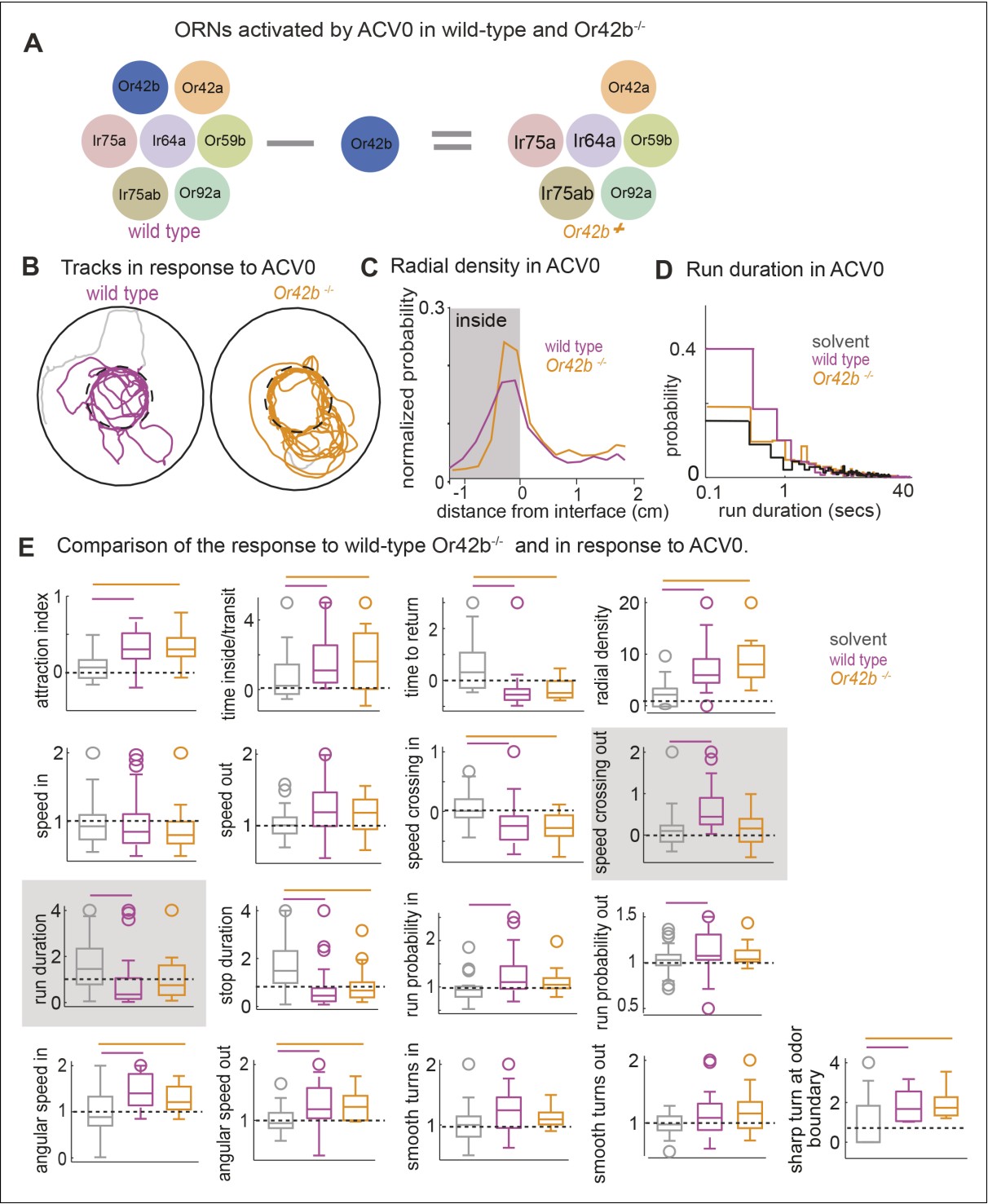

**Figure 8.** Mutating a single ORN class does not affect attraction to ACV0 but changes certain motor parameters. (**A**) In the Or42b mutant, a single ORN class is non-functional. (**B**) Sample tracks showing that both wild type and Or42b⁻/⁻ flies are attracted to ACV. Tracks also show that the mutant fly is closer to the odor interface than the wild-type flies. (**C**) Mean radial density. Although both wild type and Or42b⁻/⁻ flies are attracted to ACV, mutant flies show a sharper distribution at the interface (KS test p <0.001). (n = 29 for control, n = 25 for Or42b mutant) (**D**) Run duration is not modulated in the Or42b mutants. (**E**) 17 motor parameters of wild type and Or42b⁻/⁻ flies in response to ACV. Run duration and speed while crossing out are modulated more strongly in the wild-type compared to the mutant. Shaded box show parameters which are significantly modulated in the wild type but not Or42b mutant.

*Figure 8. continued on next page*

*Figure 8. Continued*

The following figure supplements are available for Figure 8:

**Figure supplement 1.** Responses of wild-type and Or42b mutant flies to the solvent control are not significantly different for any parameter even if not corrected for multiple comparisons.

(*Figure 9A*). The *Ir8a* mutant flies are still attracted to ACV0 (*Figure 9D*, p <0.0001 for both control and mutant in a ranksum test); there is a small but statistically insignificant decrease in attraction to ACV0 in the mutants compared to the control flies. However, there is a large change in the modulation of 5 motor parameters out of the 17 under study (*Figure 9D*). The most striking difference between the wild-type and mutant is that the immediate modulation of speed upon entering the odor-zone is completely abolished in the mutant (*Figure 9B*, p >0.1 in a ranksum test for the mutants). Similarly, modulation of speed as the fly exits the odor-zone is also abolished (*Figure 9C*). Most of the parameters affected in the mutants are those that, in control flies, are affected as the fly crosses into the odor-zone or are modulated inside the odor-zone. Thus, it is likely that the *Ir8a*-mutant strongly affects changes in motor program which characterizes a fly's behavior in the presence of high concentration of food odor. Consistent with this idea, the time spent/transit decreases in the mutant without a significant change in time to return (*Figure 9D*). The effect of Ir8a-ORNs shown above is not just on the fly's response to ACV but is also observed in response to other odors (*Figure 9—figure supplement 1*), and supports the notion that different ORN classes affect different motor parameters.

That the motor parameters inside the odor-zone are specifically affected in the *Ir8a* mutants suggests that two independent motor programs underlie a fly's response to odors: One motor program corresponds to the fly's search for the odor source; the other corresponds to the change in fly's behavior once the odor source is near. It is possible that *Ir8a*-ORNs preferentially modulates the motor program which underlie changes in a fly's search strategy from a global search in the absence of odor to a local search in the presence of high concentration of food odor. Consistent with this idea, *Ir8a* mutants spend less time in the odor-zone in the presence of ACV compared to control flies (*Figure 9D*). To further probe this issue, we plotted the time spent inside as a function of time after odor onset (*Figure 10A*). In the absence of the odor, both the mutant and the control flies spent a fifth of their time inside the odor-zone. Upon odor onset, there is a rapid increase in the time spent inside the odor-zone followed by a small decline. The initial increase in the time spent inside is similar for both the mutant and the control. However, the fractional time inside peaks faster in mutants suggesting that both mutant and control flies find the odor efficiently but the control flies stay inside the odor-zone for longer. We also plotted both the time inside/transit and time to return as a function of time after odor onset (*Figure 10B*). Time inside per transit was longer for the wild-type flies throughout the entire 3 min that the odor was on. Time to return is similar for the two genotypes. These data suggest that the change in the *Ir8a* mutants' behavior inside the odor-zone is manifested as decrease in time spent inside the odor-zone.

To investigate whether the findings above extend to conditions in which a fly usually encounters odors, we measured how wild-type and *Ir8a* mutant flies redistributed in response to a point source of ACV placed in the center of a circular arena (*Figure 10C*). Before the odor was presented, flies spent most of their time at the periphery (*Figure 10D*, left panel). In the presence of ACV the wild-type flies navigate to the odor source within seconds (*Figure 10D*, right panel). The *Ir8a*-mutant flies also navigated rapidly to the odor. But, the mutant flies dispersed from the odor source at a much faster rate than did the controls (*Figure 10E*). We measured the mean speeds in 5 radial bins and found that in the control flies, but not in *Ir8a* mutant flies, speed decreased in bins closer to the odor source (*Figure 10F*). No speed modulation was observed in the *Ir8a* mutants (*Figure 10F*). These results are consistent with a modular organization of behavior and with the fact that *Ir8a*-ORNs strongly affects a fly's behavior near the odor source but not how a fly finds that source.

## Discussion

Most studies aimed at understanding odor-modulated locomotion focus on a single strong attractant. As a result, the modulation of locomotion by a strong attractant is well understood. In contrast,

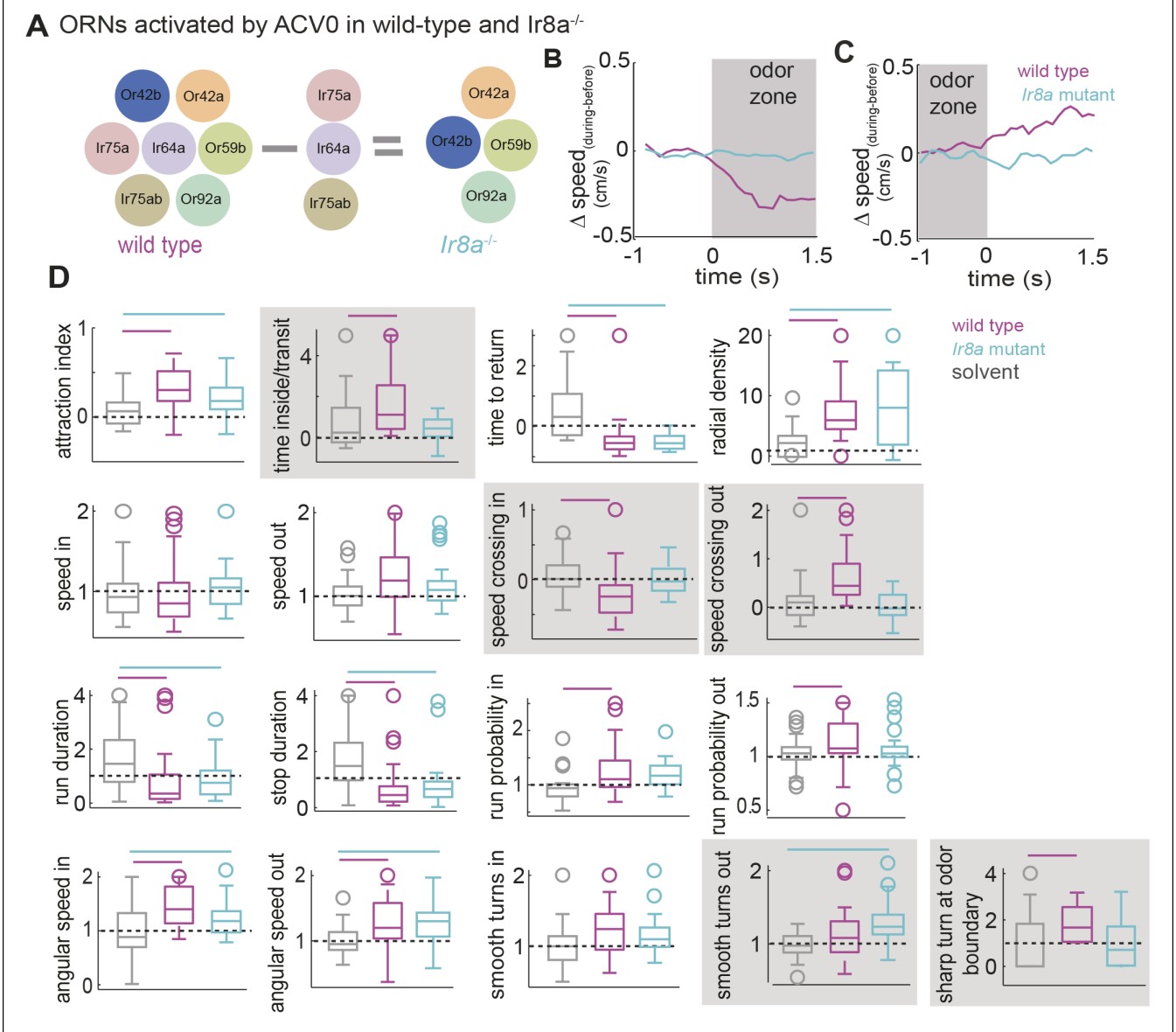

**Figure 9.** Modulation of motor programs by ACV inside the odor-zone is strongly affected in the Ir8a-mutant. (**A**) Three ORN classes activated by ACV are non-functional in Ir8a mutant. (**B**) The reduction in speed when the wild type flies enter the odor-zone in the presence of ACV is abolished in the Ir8a mutant. (n = 29 for wild type and n = 33 for Ir8a mutant). (**C**) Increase in speed when the flies exit the odor-zone is also abolished. (**D**) 17-motor parameters in the wild type and the mutant. Many aspects of a fly's locomotion are affected in the Ir8a mutant. Bars on top reflect whether the parameters are significantly different in the two genotypes compared to the solvent control. Shaded parameters are the ones affected in the Ir8a mutant.

The following figure supplements are available for Figure 9:

**Figure supplement 1.** Ir8a-ORNs directly modulate speed.

researchers have not studied the effect of diverse odors on locomotion, nor the role of different ORN classes in odor-guided locomotion. In this study, we investigated both how different odors modulate locomotion and how modulation of locomotion by a natural odor (ACV) can be decomposed into modules affected by different ORN classes. We expected a characteristic motor program underlying a fly's attraction to odor; this motor program could be recruited with different efficiency depending on how attractive an odor is. Similarly, we anticipated that mutating different ORN

classes activated by a strong attractant (like ACV) will simply reduce the attraction of the fly to that odor and thereby recruit the same motor program with less efficacy (*Figure 11A*). Our results do not support these expectations and instead support an alternate model of odor modulated locomotion in which different ORN classes affect different motor programs (*Figure 11B*); as a result, different odors with the same level of attraction can elicit very different motor programs. Implications of this model are discussed below.

## Independent control over multiple motor parameters enables flies to effectively search a complex odor environment

Two lines of evidence suggest that odors independently modulate an unexpectedly large number of motor parameters. One line of evidence is the low correlation among the 17 parameters modulated by ACV0: Of the 136 pair-wise correlations between these parameters, only 15 are significantly correlated (*Figure 4—figure supplement 1*). Consistent with this lack of correlation between motor parameters, PCA performed on the 17-dimensional behavioral description shows that 7 principal components are necessary to explain 90% of the variance in the data. Another line of evidence is that different odors modulate different parameters. BUN3 decreases stop duration but not run duration (*Figure 2*); and ethyl acetate at $10^{-4}$ dilution increases run duration but has no effect on stop duration ( *Figure 5—figure supplement 2* ). These data imply that run duration and stop duration can be modulated independently. In addition, different odors have different effects on sharp and smooth turns (*Figure 4*). Experiments with *Ir8a* mutants suggest that the acute decrease in speed observed within the first second of encountering an odor might be yet another independently controlled parameter. Thus, it appears that at least 5 parameters: stop duration, run duration, sharp and smooth turns and acute changes in speed are all modulated independently. These results together support the idea that at least 5 independently controlled parameters underlie a fly's response to odors.

Despite the fact that we are characterizing the behavior using 17-parameters, this description is unlikely to be a complete description of the behavior. A limitation of our analysis is that it does not describe the dynamics of a fly's locomotion. Simple generative models like Hidden Markov Models which has been successfully used to model locomotion of simpler animals (*Gallagher et al., 2013*), are poor fits to a fly's tracks (data not shown) because of the complexity of a fly's locomotion. Ultimately, a hierarchical statistical model (such as a Hierarchical Hidden Markov Model) is necessary to model both how a fly searches its environment and how odors modulate this search. Even without a hierarchical model, it is clear that processes which govern a fly's behavior outside the odor-zone are distinct from the processes which govern its behavior inside it. Consistent with this idea, most of the correlated parameters (*Figure 4—figure supplement 1*) correspond to the fly's behavior inside the odor-zone. Thus, a sharp decrease in speed as a fly enters the odor-zone is predictive of the overall decrease in speed, decrease in run duration and increased turning inside the odor-zone. It is likely that these 4 parameters form part of a coordinated response which constitutes a local search near high concentrations of food odor.

Most studies of odor-guided locomotion in animals are designed with the inherent assumption that the problem an animal is solving in nature is that of navigating to or away from a single salient odor. The problem of finding a distant odor source is certainly one important class of problems that the olfactory system solves. Studies across multiple species have shown that a conserved mechanism is in place to detect and track a particularly salient odor source over long distances (*Johnsen and Teeter, 1985*; *Porter et al., 2007*; *van Breugel and Dickinson, 2014*; *Kennedy, 1983*; *Cardé and Willis, 2008*). But it is unlikely that tracking an odor source over long distances is the only problem an olfactory system is trying to solve. A more general class of problems faced by the olfactory system is to navigate a complex odor environment (*Riffell et al., 2014*) consisting of multiple odor sources of varying salience. Navigating such a complex odor environment requires fine control over locomotion. It is likely that an animal's overall search strategy is under multimodal control and is flexibly linked to the overall visual, wind and odor environment. Our experiments suggest, surprisingly, that an important feature of this control is a precise modulation of motor parameters by the exact pattern of ORNs activated. This component of an animal's odor-guided locomotion has been overlooked so much so that in most experiments an odor is treated as a 'scalar' which either turns on a motor program or changes the gain of an ongoing behavior.

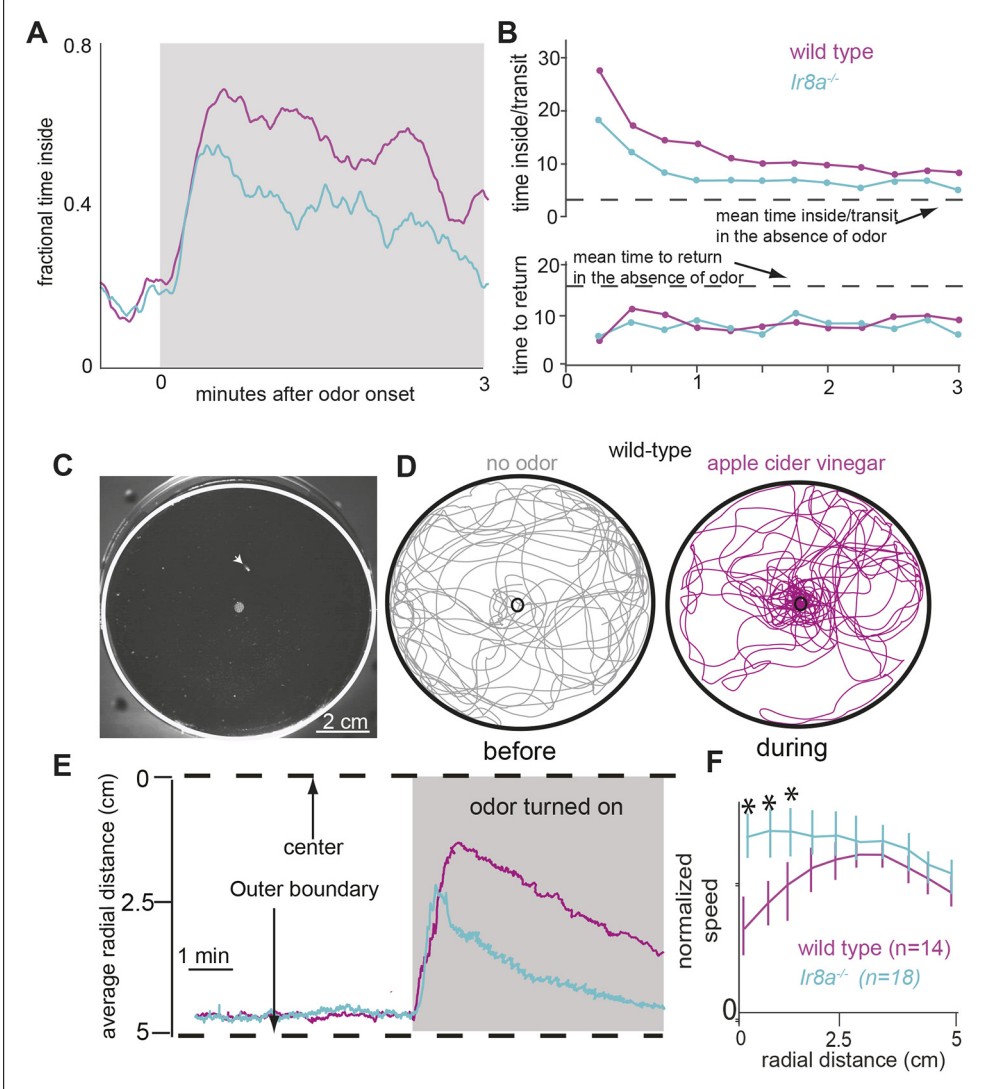

**Figure 10.** Ir8a mutants approach ACV at the same rate as wild-type but spend less time in proximity to it. (**A**) Ir8a mutant find the odor as well as the wild-type flies but because they spend less time inside the odor-zone on each visit their attraction to odor decreases with time at a faster rate than wild-type flies. (**B**). The time a fly spends inside the odor-zone on a single visit decreases at a much slower rate for wild-type than for the Ir8a mutant. In contrast, the time a fly takes to return to the odor-zone is the same for both genotypes. (**C**) A photo of the arena with the outer edge marked with a white line. Fly can be seen as a tiny white object (marked with an arrowhead). The hole in the center is used to deliver odors. (**D**) Tracks of a control fly shows that it is strongly attracted to apple cider vinegar. (**E**) Radial density averaged over multiple flies show that Ir8a mutants find the odor as quickly as the wild-type but leave the odor much faster. (**F**) Speed (between 0—2 min after odor on) near the odor source decreases in wild type but not in the Ir8a mutants.

## Relation between ORN activity and behavior

There is general consensus that a combinatorial activation of multiple ORN classes drives behavior; however, there is little consensus regarding the role of a single ORN class in behavior and the principles by which activities from multiple ORN classes are combined to yield behavior. For instance, in one study it was shown that the *Or42b*-ORNs plays an important role in attraction to a natural odor (*Semmelhack and Wang, 2009*). In contrast, another study finds that Or42b-ORNs (*Knaden et al., 2012*) does not contribute to attraction at all and proposes a more modest contribution of a single ORN class to behavior. Consistent with the study in *36* we show that activation of *Or42b*-ORNs is not enough to attract flies to an odor. Similarly, *Or42b* mutants are still strongly attracted to ACV

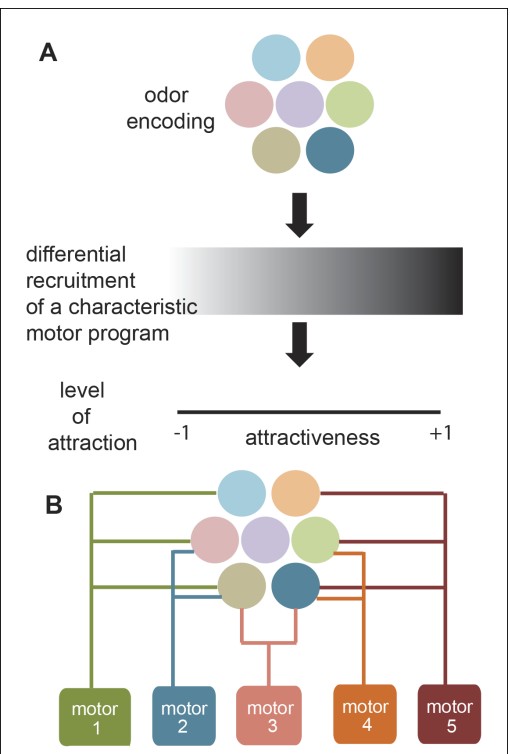

**Figure 11.** A new framework for olfactory behaviors. (**A**) Current framework. Based on the pattern of ORNs activated by a given odor, a stereotypical motor program is activated with different efficacies leading to different levels of attractiveness. (**B**) Novel Framework. Different motor parameters are modulated by overlapping sets of ORNs leading to an odor dependent redistribution of the fly.

suggesting a modest role for *Or42b*-ORNs in mediating attraction to odor. These differences in behavioral response to Or42b-ORNs in different labs could result from differences in behavioral paradigm or the state of the fly. Apart from *Or42b*-ORNs, we have also performed experiments in which we specifically activate *Or42a*-ORNs, *Or59b*-ORNs and *Or7a*-ORNs but failed to observe strong attraction. Thus, it is unlikely that activating a single ORN class will result in the strong attraction observed in the presence of a natural attractant. On the other hand, most odors which activate multiple ORN classes are somewhat attractive to a hungry fly suggesting that a mild attraction to odor can be elicited readily and does not require a unique or rarely activated combination of ORN classes. Similarly, attraction to natural odors is robust: even the *Ir8a* mutant is attracted to ACV. These findings together suggest that there are many redundant paths to odor-mediated attraction.

More importantly, our work suggests a novel framework for the study of relation between ORN activity and behavior. Although, activation of *Or42b*-ORNs by themselves does not elicit attraction, they have a significant effect on some motor parameters but not others. Another ORN class, *Or42a*-ORNs, affects another subset of motor parameters. These data suggest that individual ORN classes only affect a subset of motor parameters. Our experiments with different ACV concentrations show that as the ACV concentration is increased, more ORN classes are recruited which in turn result in the modulation of more motor parameters. These data led us to propose a fundamentally different model of odor-guided locomotion in which individual ORN classes affect a small number of motor parameters (*Figure 11B*). The above model finds strong support in experiments aimed at dissecting the contribution of different ORN classes to the fly's response to ACV. We used three complementary approaches to activate subsets of ORNs: activation of known ORN classes using low concentrations of odors, different ACV concentrations and different OR mutants. We find that a given pattern of ORN activates the same motor parameters irrespective of how the ORNs are activated: activation of *Or42b*-ORNs by ACV4 or ETA8 decreases run duration and increases turning. Similarly, ACV does not elicit a decrease in speed until the concentration is high enough to activate Ir8a dependent ORNs; this observation is consistent with the strong effect of Ir8a-ORNs on speed. Overall, activation of different subsets of ORN which are all activated by ACV results in the modulation of different subsets of motor parameters.

Multiple lines of evidence suggest that *Or42b*-ORNs affect run duration. Similarly, Ir8a-ORNs strongly affects the modulation of speed. It is tempting to conclude that different motor parameters are simply modulated by instantaneous summed activity from a subset of active ORN classes. This model is unlikely to be correct for all motor parameters. In particular, the principles underlying modulation of both sharp and smooth turns appear to be impervious to this simple analysis. This is reflected in the fact that although Or42b-ORN activation can affect angular speed, modulation of angular speed is not significantly affected in the *Or42b* mutants. Moreover, the facts that BUN3 modulates sharp turns but with little spatial specificity; and, ACV modulates it with great spatial specificity which is increased in the *Or42b* mutants indicate that a more complex rule than simple summation of ORN activities is at work. In sum, our work indicates that each motor parameter is

modulated by a subset of ORNs; but, the relation between ORN classes and motor parameter could be simple for some parameters and complex for others. In the same vein, in this study in deriving the relation between ORN activation and behavior we have focused on which ORN classes are active. Future experiments will determine whether the level of activation and temporal characteristics of ORN spike response is relevant to behavioral output. To delineate the relationship between the spatio-temporal pattern of ORN activity and motor parameters, we need both a hierarchical model of a fly's locomotion and experiments aimed at understanding how inputs from different ORNs are integrated by higher-order neurons in the fly's brain.

Our experiments with the *Ir8a* mutant strengthen our conclusion regarding modular organization of olfactory control on locomotion, and also suggest one organizing principle. Removal of the three Ir8a dependent ORN classes reveals strong effect on some motor parameters while leaving other motor parameters completely unaffected. The motor parameters affected by *Ir8a* mutants strongly affect a fly's behavior inside the odor-zone without strongly affecting its behavior outside the odor-zone. These results further support the idea that the modulation of locomotion which leads to attraction to the odor source and modulation of locomotion which results in a fly's exploration of regions of high odor concentration are controlled independently. Our data suggests that activation of *Ir8a*-ORNs is important for initiating a local search characterized by short slow runs and frequent but short duration stops. This local search represents a part of an overall change in strategy that promotes a thorough search of a local area. Recent work has demonstrated that locomotion and proboscis extension to initiate feeding are antagonistically related (*Mann, et al., 2013*). Thus, food odors and tastants could potentially work in concert to put brakes on locomotion and initiate feeding.

## Implications for flexible control of behavior

An animal's behavioral response depends not only on the immediate sensory stimulus but also on context and the animal's goal. There are two competing models which account for flexible sensorimotor transformation. In the first and the dominant model, sensory processing creates an internal representation of the external world. Using this internal model, decision-making circuits can take a behavioral decision which is then executed by motor circuits. A student of neuroscience would be strongly inclined to this model after picking up any neuroscience textbook (*Kandel, 2000*; *Purves, 2008*). The strength and the allure of this model lie in the access to an internal representation. In this model, the versatility in animal behavior arises by endowing the nervous system with cognition - the ability to plan actions based on an internal model of the external world (*Cruse, 2003*; *Turvey and Fonseca, 2009*). This model has a centralized 'executive system or circuit' which takes into account sensory input and the state of the animal to plan action (*Cruse, 2003*). But, efforts to create robots with such a subsystem have failed (*Brooks, 1991*). Updating internal representations with every small act is just too slow.

The alternative model proposes a modular organization containing parallel sensorimotor loops, each of which represents a solution to one aspect of an ecological problem (*Wehner, 1987*; *Wessnitzer and Webb, 2006*). In this model, the role of sensory systems is not to create an internal representation of the world but to extract behaviorally relevant features in the environment. A presumed limitation of the modular organization is that it would result in simple, stereotypical behaviors and hence this framework is considered relevant only for simple behaviors performed by simple organisms. Recent behavioral work has revealed that even animals with simple nervous systems are capable of complex behaviors. Fruit flies, for instance, can make use of both idiothetic (*Pick and Strauss, 2005*) and allothetic cues to navigate and also exhibit spatial memory(*Ofstad et al., 2011*). At the other extreme, recent work in the mammalian retina has revealed that many retinal ganglion cells are feature detectors and only affect specific behaviors (*Zhang et al., 2012*). These studies have led to the realization that a modular organization is not necessarily inconsistent with complex behaviors or complex nervous systems.

We propose a model for flexible sensorimotor transformation in the olfactory system which was proposed in the context of work in primate vision. This model suggests that perception (or internal models) of objects occur via circuits that are independent from action directed at the same object (*Goodale and Milner, 1992*). As proposed in vision, there are two separate computations – one for action and one for perception for every sensory system. The system for action, simple and complex, allows rapid online control. The system for perception is relatively slow and exerts control over

action over a longer time course and modulates the system for action based on the state of the animal and its goal.

Superficially, a modular system like the one we propose appears hardwired. But in reality, by modulating the relation between ORN activation and change in motor parameter one can flexibly couple different motor output to the same sensory input. The implication of our model is that at any instant, based on its assessment of its state and goal, an animal can rapidly transform sensory input into behavioral output. This study describes the sensorimotor transformation for an average hungry fly over a 3 min exposure to odor. An important feature of this transformation is that different ORN classes affect different motor parameters. But, the effect of activating a given ORN class on a motor parameter is likely to change depending on other variables such as the state of that individual and sensory context. *Figure 10* shows an example of how the behavioral response changes with continuous exposure to odor. Future studies will determine whether this modulation implies a change in weighting between ORN spikes and a given motor parameter or a complete reorganization between ORN activity and locomotor output.

## Materials and methods

### Flies

Control flies were either $w^{1118}$, laboratory cultures of 200 wild caught Drosophila melanogaster used previously by other labs (*Ofstad et al., 2011*; *Bhandawat, 2010*) or $Or42b^{+/-}$ heterozygotes. Or42b-mutant flies ($Or42b^{EY14886}$) were obtained from Bloomington stock center and backcrossed to $w^{1118}$ for 10 generations. Flies used for the behavioral assay were raised in 'sparse culture' condition as described previously (*Bhandawat, 2010*). Briefly, 100~~150 eggs were collected in a standard corn-meal-agar media bottle and left in the 25o°C incubator with 12 hr:12 hr light:dark cycle until the adult flies eclosed. Newly eclosed flies were transferred to fresh food vials. One day before behavioral experiments, 10~20 flies were transferred to an empty vial containing a wet paper for starvation.

### Custom-built behavioral arena for ring assay

The custom-built behavioral arena (details in *Figure 1—figure supplement 1*) was connected to an odor-delivery system that was similar to the one described previously (*Bhandawat, 2010*). The construction of our behavioral arena is diagrammed in *Figure 1—figure supplement 1*. The significant features of the arena are described below. A sharp interface between the odor-zone and no odor-zone is an important feature of the arena. The sharpness of the interface largely resulted from the vacuum being run at a much higher rate than the air flow thereby sucking air radially inwards into the arena. Vacuum pulled 6 liters/min while the flow through the air tube was 1.2 liters/min. To allow for a radial inward flow of air, we machined holes into the outer rim.

Air flow rates were kept low to minimize anemotactic responses. The airflow at the interface of air tube and the arena is 0.07 m/s and the highest speed anywhere in the arena is 0.11 m/s. These speeds aretwofold lower than the lowest speeds used to induce anemotaxis in flight (*Budick et al., 2007*). Consistent with these low flow rates, we saw little effect of turning on the air on the fly's behavior (*Figure 1—figure supplement 3*).

In constructing the junction between the air tube and the upper plexiglass plate, there were two important considerations. First, the joint had to be as clear as possible to minimize occlusion of the fly's image on the camera. Second, the edge of the tube had to be flush with the upper plate so that the fly did not feel a significant mechanosensory edge. We also machined a thin Mylar sheet with 0.5 mm holes separated by 1 mm. The air/odor entered the arena through this Mylar sheet. The flies did not show a strong preference for exploring the junction between the air tube and the plate or the Mylar sheet.

To enhance the visual contrast between a fly and the background, the bottom plate and mesh were sprayed black and the vacuum bottle was covered with black tape. Infrared LED lights were placed around the arena to achieve uniform illumination. As much as practicable, we covered or painted any reflective surface to prevent spurious reflections.

## Odor delivery and odor preparation

The olfactometer we used was same as the one previously described (*Bhandawat, 2010*). The background flow was set at a rate of 1 liter/min. The flow through a secondary air stream at 200 ml/min was controlled by a valve which regulated whether the air passed through the odor vial or not (*Figure 1A*). Ethyl acetate (CAS # 141-78-6), 2-butanone (CAS # 78-93-3) and ethyl butyrate (CAS # 105-54-5) (Sigma-Aldrich) were serially diluted in paraffin oil (Baker). Organic apple cider (Spectrum Naturals, Filtered) vinegar was diluted in distilled water.

## EAG measurements in the arena

We used EAG to quantify odor concentration in our behavioral arena. To measure EAG, we inserted a glass electrode into the fly's eye. After inserting the electrode, we used a UV-cured glue (KOA-300 from Kemxert Corporation) to glue the electrode to the eye. This electrode served both as a tether and as a ground electrode for sensillum recording. This electrode was mounted on a micromanipulator to position the fly at different places in the arena. Next, another electrode (a sharp electrode which is used to perform sensillum recording) was inserted into the fly's antennae. This second electrode was also mounted on a micromanipulator. This whole ensemble was moved around the arena. For each measurement, the electrode was held stationary at a given location and the odor was turned on. The ensemble must be horizontal and should also travel horizontally. For a given experiment, the electrode needs to be stable for all measurements. This is because the value of EAGs strongly depends on the place and depth of insertion. Each measurement was repeated 4 times a given location. Because absolute EAG values vary widely from experiment-to-experiment, the values were normalized by dividing by the peak response obtained inside the odor-zone. The normalized response in *Figure 1D* is obtained by integrating over a 15 s window after odor on. This window was chosen because the response reaches a plateau within 15 s.

## Single sensillum recording and patterns of ORNs activated by ACV

ORNs were recorded using single-sensillum recording as previously described (*Bhandawat et al., 2007*). Briefly, flies were immobilized in the trimmed end of a plastic pipette tip under a 50X air objective mounted on an Olympus BX51WI microscope. A reference electrode filled with saline was inserted into the eye, and a sharp saline-filled glass capillary was inserted into a sensillum. Recordings were obtained with an A-M Systems Model 2400 amplifier, low-pass filtered at 2 kHz and digitized at 10 kHz. ORN spikes were detected using routines in IgorPro (Wavemetrics). Recorded ORNs were matched with ORN class based on: (1) sensillum morphology and size, (2) sensillum position on the antenna, (3) spike amplitude, (4) spontaneous spike frequency, and (5) odor tuning of cells in a sensillum. Because all these properties are a stereotyped function of cell lineage, together they form an unambiguous signature of ORN identity (*Bhandawat et al., 2007*; *de Bruyne, 1999*; *de Bruyne et al., 2001*).

Spike times were extracted from raw ORN recordings using routines in Igor Pro. Each odor was presented 4-–6 times at intervals of 40–60 s (a 'block" of trials). The response to the first presentation was not included in our analysis. Each of the remaining trials was converted into a peri-stimulus-time histogram (PSTH) by counting the number of spikes in 50 ms bins that overlapped by 25 ms. These single-trial PSTHs were averaged together to generate a PSTH describing the response to an odor in a given experiment. Multiple cells corresponding to each ORN class and each cell were tested with a given odor in multiple experiments, each with a different fly. Average PSTHs represent the mean ± s.e.m computed across experiments.

To obtain the pattern of ORNs activated by different concentrations of ACV, we performed single-sensillum recording from all basiconic and coeloconic sensilla. Single sensillum recording was performed as previously described (*de Bruyne, 1999*; *Bhandawat et al., 2007*; *Olsen and Bhandawat, 2007*). Sensillum type were identified first by morphology and a panel of diagnostic odors. We measured the response of each sensilla to the highest concentration of ACV. ORNs which did not respond to the highest concentration in three different flies were deemed unresponsive to ACV. We repeated this process at each lower concentration of ACV until at ACV4, only Or42b-ORNs respond to ACV. This was confirmed by showing that the EAG response to ACV4 is abolished in the *Or42b* mutant.

## Details of the behavioral procedure for ring assay

Flies were starved for 18~~24 hr before the behavioral assay. A single female fly was cold-anesthetized and placed in the behavioral arena. We usually waited for 30 min to give flies time to acclimatize to new environment. Although experiments were performed during the peak circadian activity periods, there were still some inactive flies and we discarded ones that did not walk for at least half of the acclimation period. Individual flies walked either on the floor or the ceiling of the arena without many transitions ( <1 on average) between the two. As a population, flies had equal probability of being on the floor or the ceiling. Each trial consisted of 3 min of no odor (before), 3 min of odor (during) and 3 min of no odor periods (after). Behavior videos were acquired at a rate of 30 frames per second and processed offline to obtain the position of a fly using a custom-written MATLAB script (available at https://github.com/bhandawat/fly-walking-behavior). During the behavioral assay, we waited for at least 3 min between different odor trials to avoid any effects caused by residual odor from the previous trial. We tested multiple odors for a single fly and used odors such as apple cider vinegar to which a fly has a strong behavioral response at the end of the experiment.

## Image processing

The air tube made it impossible to image the arena with a single camera because images behind the air tube were severely distorted (see *Figure 1B*). Instead, we imaged the arena with two cameras. The un-occluded sides of the image from the two cameras were registered and stitched to obtain an un-occluded view of the arena (see videos). The resulting video was used to extract a fly's position. We subtracted the average frame from each frame to obtain a background-subtracted video in which the fly is the only bright object in an otherwise dark background. The position of the fly was obtained as the centroid of the largest bright object (usually the only bright object) in a frame. When the fly was directly under the edge of the air tube, background-subtracted video showed two separate objects instead of one, which caused errors in tracking. To correct these errors, the centroid was calculated as the center of both objects when the fly was near the edge of the air tube. We checked auto tracking by measuring the fly's instantaneous velocity. Frames on which the instantaneous velocity exceeded 3 mm/s represented frames where the fly's position had been incorrectly assigned. These errors in tracking were less than 1% and they were manually corrected. Lastly we transformed the track such that it represents the view from a single camera placed directly above the arena. All image processing was performed in MATLAB using the Image Processing Toolbox.

## Definition of odor periods and odor boundary

Flies enter the odor-zone at variable times after the odor is turned on. Thus, there are multiple reasonable ways to delineate 'before', 'during' and 'after' period. We chose the time between the start of the trial to the nominal time when the odor was turned on as the 'before' period. Therefore, the 'before odor' period always had a fixed length of 3 min. The definition of 'during odor' period depended on whether the fly was inside the odor-zone when the odor was turned on or outside. If the fly was inside the odor-zone, the 'during' period started at 3 min and 5 s because based on the EAG measurements it took 5 s for the odor to reach the arena. If the fly was outside the odor-zone, the time at which it first entered the odor-zone marked the start of the 'during' period. Thus the length of 'during' period was variable and <3 min long. The 'after' period started 6 min after the start of the video and lasted till the end of video acquisition.

From the EAG measurement (*Figure 1C,D*) and smoke test (*Figure 1—figure supplement 2*), we determined that there was a sharp odor boundary at the physical inner rim (1.2 cm away from the center of the arena) and we estimated that there was no odor response at 1.5 cm from the center of the arena. Moreover because we used the fly's centroid (typically on the thorax or abdomen of a fly) instead of actual antenna location, the fly could be at 1.65 cm (adding 0.15 cm for the half-length of the fly) facing radially inwards and still be experiencing odor. In the presence of ACV, flies often stopped just before entering the odor-zone. For ACV at $10^{-2}$ and $10^{-1}$, these stop events occur within 1.65 cm of the arena center. In case of pure ACV, many flies stopped outside 1.65 cm, so we assumed that centroid locations at this point was a better measurement of the actual odor boundary. We found this to be at 1.9 cm away from the center of the arena and used it as a conservative estimate of the odor boundary.

## Track smoothing and run analysis

The raw coordinates of the fly were smoothed using a sliding window which was 10 frames long (0.33 s). The window size was chosen empirically by comparing the raw and smoothed tracks.

To convert camera pixels into real world distances, we measured the diameter of the arena (6.4 cm) in pixel units to obtain a conversion ratio of 1 pixel to 0.1 mm. The typical size of a fly in a video was about 30 pixels which corresponded correctly to the actual size of flies we used (~~3 mm). Speed was measured from the smoothed coordinates by simply dividing the displacement in consecutive frames by the time between them (0.033 s).

Runs and stops were assigned using the Schmidt trigger algorithm already introduced in earlier studies (*Martin, 2004*; *Robie et al., 2010*). If the instantaneous walking speed of a fly was lower than 0.5 mm/sec, then the fly was stationary or at a stop and if the speed was faster than 1 mm/sec, then the fly was moving or on a run. If the speed was in between those two thresholds, it was classified as a continuation of the previous run or stop. Comparison of automated run-stop analysis with manually scored run-stop results showed a 100% agreement except for two systematic biases. First, very short stops lasting 1--3 frames or less than 100 msec went undetected by our algorithm. Second, the algorithm is 1–3 frames late in recording the start and ends of runs.

Angular speed was measured by calculating the curvature of the smoothed tracks during runs. To minimize errors occurring during fly's slow movement, we disregarded angular speed changes when flies were moving slower than 2 mm/sec. In our experiment, flies have at least two qualitatively different modes of turning: they can either turn smoothly over hundreds of millisecond or sharply over much shorter time. To parse the fly's turns into these two modes, we divided the turns into 'smooth turns' or 'sharp turns'. A fly was designated as performing a turn when the sum of it angular velocity over 5 frames was >0.3 radian. If the sum was greater than 1.3 radians, it constituted a sharp turn (*Figure 12*).

## Details of procedure and analysis for behavior with a point source of odor

Behavioral studies with a point source of odor were performed in an arena that was a 10 cm in diameter and 4 mm in height. A hole with 1.5 mm diameter was drilled in the middle of the arena to create an odor delivery port. Odors were delivered by attaching an odor vial below the hole. Odors entered the arena passively, i.e. there was no airstream carrying the odor. The chamber was stabilized with magnets which held the chamber firmly in place against a metallic ring stand. Experiments were performed in dark under infrared illumination. A camera was placed 25 cm above the chamber. The chamber was painted black; a fly was the only bright object. Tracking was performed as in the ring assay.

Preparation of the fly cultures, starvation and other details are as described for the ring assay in section 5 above. The behavior was performed one fly at a time to prevent distortions due to flies bumping against each other. Flies were tracked for 5 min before odor onset and 5 min during the odor period. 5 min was sufficient time for most flies to venture into the center even before odor onset. This was critical for assessment of statistics.

To calculate the radial density, we simply averaged the instantaneous radial position and averaged over all the flies to obtain the radial density as a function of time. To calculate speed as a function of radial distance, we divided the arena into 10 concentric bins and measured the mean speed in each bin. The resulting distribution was normalized for individual flies by dividing by the maximum speed. The mean and standard deviation over flies is reported in *Figure 10F*.

## Behavioral analysis and selection of motor parameters

In conventional olfactory behavioral assays such as a T-maze test, results are typically represented as a performance index, which measures the proportion of time that flies spend inside the specifically defined odor zone (reviewed by Davis 2005). We used a parameter similar to the performance index, 'attraction index', as a measure of the relative change in time a fly spends inside the odor zone between before and during the odor stimulation periods.

We used the fly's response to apple cider vinegar to design an appropriate parameterization of the fly's odor-evoked behavior. Underlying the fly's attraction to apple cider vinegar, both its time to return to the odor zone and time spent inside changed. This implies that the change in fly's behavior is not limited to the time that a fly is actually in the presence of odor (i.e. in the odor zone).

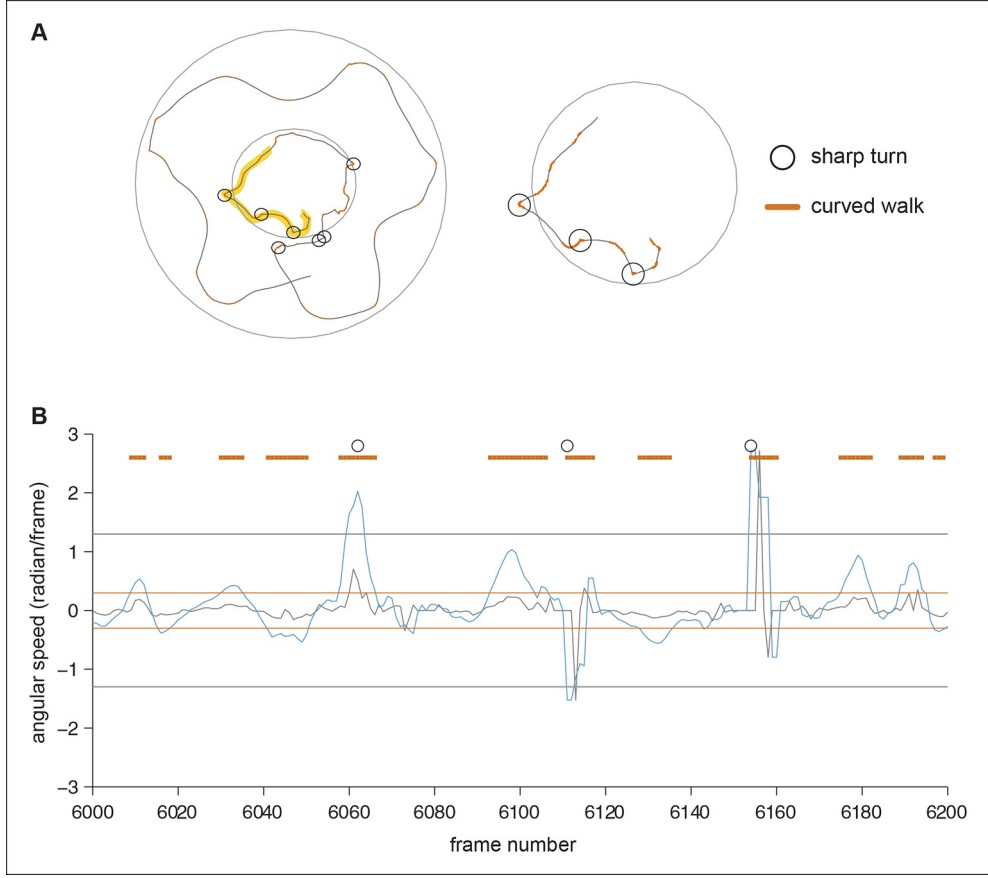

**Figure 12.** Determination of curved walk and sharp turns. (**A**) A part of a fly's walking trajectory. The curved walk is indicated with orange line and sharp turns are marked with black circles. The right panel shows the magnified walking trajectory from the left (marked by yellow highlight). **B**) The angular speed (grey line) and sum of 5 frame-long angular speed (blue line) for the frames corresponding to the highlighted track above. The orange lines mark the threshold for the curved walk and the grey lines mark the threshold for sharp turns. Frames detected as curved walks (orange) and sharp turns (black circle) are marked.

We also observed that a fly's speed increases outside the odor zone in the presence of odor and decreases inside. Thus, the same parameter can change in different ways outside and inside the odor zone. Therefore, we investigated the change in a given parameter both inside and outside the odor zone. In addition, we observed that flies sometimes showed acute behavioral changes on crossing the odor boundary, so we added certain parameters to depict the acute changes (e.g. speed crossing in).

## 17 Motor parameters

Below, we describe each parameter in the 17-dimensional behavioral space. Two considerations influenced our statistical approach. First, the modulation of many parameters depended on the value of that parameter before odor onset. For instance, if one fly repeatedly went inside the odor-zone before odor onset, it would likely do so even when the odor is turned on. To account for these intrinsic differences in a fly's tendency, we are reporting normalized parameters. Second, modulation of a given motor parameter by an odor can result from either the odor itself or the solvent control. Therefore, to establish whether a given parameter is modulated by a given odor, we will perform statistical tests between the distributions of a given parameter in the solvent control versus its distribution in the presence of the odor.

1. **Attraction index** = $\frac{(Time\ spent\ inside\ in\ the\ during\ period) - (Time\ spent\ inside\ in\ the\ before\ period)}{(Time\ spent\ inside\ in\ the\ before\ period)}$. This computation was performed over individual flies.

2. **Time spent/transit**: Flies entered the odor-zone multiple times in each odor period. We calculated median time inside/transit for each period. The statistic plotted was calculated as follows: $time\ inside\ per\ transit = \frac{median\ time\ per\ transit\ 'during'\ --median\ time\ per\ transit\ 'before'}{median\ time\ per\ transit\ 'before'}$

3. **Time to return**: This was calculated in the same manner as Time spent inside/transit except that the median time outside/transit was used instead of median time inside/transit.

4. **Radial density**: In addition to sorting fly's location as 'in' and 'out' of odor zone, we further analyzed the fly's location as distance from the center of the arena. We binned the data into 12 bins and normalize the fraction of time spent in each bin by area of each bin.

5. **Speed inside** = $\frac{(average\ speed\ inside\ 'during'-average\ speed\ inside\ 'before')}{average\ speed\ inside\ 'before'}$

6. **Speed outside** was calculated in the same way as speed inside except with average speed outside.

7. **Speed crossing inside**. Each fly crossed into the odor-zone multiple times. Each crossing event was aligned with the time the fly crosses inside as 0. An average speed trace was obtained for every fly for the 'before' and 'during' period. The speed crossing inside in a given period was calculated as the average speed in a window 1 s to 2 s after the fly enters the odor-zone. The statistic we used was calculated as follows: $Speed\ crossing\ inside = \frac{speed\ crossing\ inside\ 'during'-speed\ crossing\ inside\ 'before'}{speed\ crossing\ 'before'}$

8. **Speed crossing outside** was measured in the same way as speed crossing in except that crossing events which took the fly outside from inside was used instead of from in to out.

9. **Run duration.** For each period and fly, the median run duration was calculated. The plotted run duration statistics was calculated as Run duration = $\frac{run\ duration\ 'during'}{run\ duration\ 'before'}$

10. **Stop duration** was calculated in a manner analogous to run duration.

11. **Run probability in.** Run probability inside was calculated as the ratio between the probability that a fly is moving during odor period and before odor period calculated when the fly is inside the odor-zone.

12. **Run probability out** was calculated in the same way as run probability inside except that we used the values when the fly was outside the odor-zone.

13. **Angular speed inside** was calculated in the same way as speed inside (parameter 5).

14. **Angular speed outside.** Same as 13 but for the times when the fly is outside.

15. **Smooth turns in.** Using the algorithm described above, we calculate the frames in which the fly is turning but not making sharp turns. This parameter plots the fraction of frames during which the fly is performing smooth turns.

16. **Smooth turns out.** Same as 15 but outside the odor-zone.

17. **Sharp turn at boundary.** We measured the fraction of sharp turns which took place at the odor border, that is in a ring 2 mm around the border.

## Other behavioral analysis not covered by the 17-parameters above

a. Traces in *Figure 2a,b,3b,9b,c* which show how speed changes after crossing inside or outside the odor-zone were obtained as follows. We calculated speed in 160 ms bins (5 frames in a 30 frames/s video). For each fly we obtained the average speed trace in the before period. Next, we obtained an average of average to obtain the change in speed before odor onset. Using a similar procedure, we obtained the trace during odor onset. The plotted traces were obtained by subtracting the during-trace from the before-trace.

b. Stop duration, run duration and run speed histogram (2D-–F) are simply distributions over all episodes of each.

c. Fraction of turns plot in *Figure 2H* is obtained as described for speed in a.

d. Radial density plots in *Figure 8C* were obtained by averaging the radial densities across all flies.

e. Fractional time inside as a function of time (*Figure 10A*) was obtained by evaluating what fraction of flies was inside the odor-zone in any 5 s period. Time inside/transit and time to return were calculated similarly but binned over 15 s.

## Statistical analysis

Most parameters did not distribute normally. Wherever a measure of central tendency was necessary we have used medians. Because of non-normality, boxplots are used to represent the data. On each box, the central mark is the median, the edges of the box are the 25th and 75th percentiles, the

whiskers extend to the most extreme data points not considered outliers, and outliers are plotted individually. In choosing the 17-dimensional space, we used p <0.05 as a criteria for considering a given parameter. To ascertain whether a given parameter is significantly modulated by odor, we corrected for multiple comparisons by applying the Bonferroni correction and used a p $<(0.05/17)$ =0.003.

### Canonical variate analysis (CVA)

CVA is similar to PCA but explicitly designed to find linear combinations of variables which maximize the difference between groups. It does so by looking for the eigenvalues and vectors of the between-groups covariance matrix 'in the metric of' the within-groups covariance matrix. If $B$ is the within-group covariance matrix and W is the within-groups covariance matrix, then the canonical variates are obtained by solving the equation $(B^{-1}W)$ a = 0 or an eigen decomposition of $W^{-1}B$. Unlike PCA, the number of dimensions does not equal the number of variables. Rather the number of dimensions is one less than the number of groups. For comparison between two groups, all the 'variance' is captured in a single dimension, leading to simple one dimensional representation we have used in *Figures 4,5* and Supplementary *Figure 6*. To show that the separation between flies' behavior to two odors along the first canonical variate is not just an artifact of over-fitting, we randomized the odor label 50000 times and calculated the median distance between odor representations in each iteration. To assess significance, we measured the fraction of randomized trial with median > original median.

## Acknowledgments

We would like to acknowledge Cynthia Hsu and Yangning Lu for building the initial versions of the behavioral chamber. Two high school students, Tim Qi and Sidney Lasanza performed the experiments described in *Figure 10*. We would like to thank members of Bhandawat lab for discussions. Rachel Wilson, Gaby Maimon, Quentin Gaudry, Coby Schal, Sheila Patek, Richard Mooney and Christine Drea provided critical feedback on earlier versions of the manuscript. This research was partially supported by NIH NRSA Institutional Research Training Grant to SJ (5T32NS051156).

## Additional information

### Funding

| Funder | Author |
| --- | --- |
| Duke University | Vikas Bhandawat |
| National Institute of Neurological Disorders and Strok | Seung-Hye Jung |

The funders had no role in study design, data collection and interpretation, or the decision to submit the work for publication.

### Author contributions

SHJ, Conception and design, Acquisition of data, Analysis and interpretation of data, Drafting or revising the article; CH, Acquisition of data; VB, Conception and design, Analysis and interpretation of data, Drafting or revising the article

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
