## [Decision Letter]

Thank you for submitting your work entitled "Odor-identity dependent motor programs underlie behavioral responses to odors" for peer review at *eLife*. Your submission has been favorably evaluated by K VijayRaghavan (Senior Editor) and two reviewers. Both reviewers have agreed to reveal their identities: Mani Ramaswami (Reviewing Editor and Reviewer 1) and Ron Calabrese (Reviewer 2).

The reviewers have discussed the reviews with one another and the Reviewing Editor has drafted this decision to help you prepare a revised submission.

In "Odor-identity dependent motor programs underlie behavioral responses to odors," Jung et al. make the valuable point that "attraction" and "repulsion" are inadequate measures of the effect of an odorant. Thus, two "attractive" odorants, apple cider vinegar (ACV) and 2-butanone at 10^-3^ dilution (BUN3), trigger responses and motor patterns that are quite different from one another. Using a clever behavioral arena in which they are able to create sharp odor boundaries they measure multiple (largely independent) parameters of locomotion that contribute to odor 'attraction behavior' (seeking and maintaining contact with attractive odors such as those in foodstuffs). The system allows for the stimulation and genetic elimination of specific ORNs and thus allows a powerful dissection of behavior. They find that rather than a simple relationship between a specific ORN class and attraction behavior a more complicated picture emerges that suggests a modular organization of olfactory guided attraction. Specifically they find that in response to odors, flies modulate a surprisingly large number of motor parameters, similarly attractive odors elicit changes in different motor parameters, and different ORN classes modulate different subset of motor parameters. The experiments are thorough, well controlled, and appropriately analyzed with relevant statistics.

Findings described in the paper greatly nuance and deepens our thinking about olfactory responses, going beyond simple valence arguments. In addition, it hints at a wide range of future experiments to connect specific olfactory modalities to different components of adult motor circuits. This also provides a common neural circuit framework in which think about responses to sex pheromones and non-pheromonal odorants. The data are analyzed with appropriate statistical methods.

However, several revisions are strongly recommended, largely to focus the paper on its key and simple message regarding the complexity and diversity of motor responses to odorants, to clarify the presentation and some details. One major suggestion is to present the interesting data, which are not easily consistent with Semmelhack and Wang's conclusion that Or42b-ORNs mediate ACV attraction. However rather than attempt to strongly argue against their conclusion (readers can make up their own mind based on the data presented), to use these data simply to highlight the specific contribution of the Or42b cells to the complex ACV behavioural responses measured here. A second suggestion is to carefully edit the figures, labels and legends with a view to increasing clarity and ease of understanding.

Specific comments:

1) The studies of "single-ORN" induced motor responses are useful in that they demonstrate specific motor responses to very sparse olfactory input. The point that, ACV responses are not the same as those induced by low ethyl acetate concentrations (EA7 and EA8) which activate Or42b is clear and well made. However, the conclusion that Wang and colleagues were wrong to ascribe specific ACV responses to Or42b-ORNs is more difficult to make in rigorous fashion.

A) The authors state that "Or42b-ORNs are the only ones activated at these concentrations" and cite previous publications on this point. However, it is always going to be difficult to be 100% certain that all EA7/8 responses are mediated though Or42b. Could previous experiments have lacked the sensitivity to detect small but significant responses in other ORNs? Could EA also activate other unknown sensory cells in other parts of the fly's anatomy (as appears to be the case for Benzaldelyde)? There is no data in this paper to show that flies lacking OR42b receptors do not respond at all to EA7/8 in the current behavioural paradigm.

B) Could there be more detailed information encoded in the finer aspects of the ORN response – its rate of onset, the spike frequency or rate of change of spike frequency – that allow a fly to discriminate between Or42b signals stimulated by EA and ACV?

C) The above two points are partially (and quite strongly) argued against by analysis of the ACV response of flies lacking the Or42b receptor, but again this does not reproduce the experiment by Semmelhack and Wang, who used shits-based silencing of Or42b-ORNs. There are scenarios in which the two different modes of "silencing" can lead to different results. In this regard, "silencing" may not be the correct term for what is accomplished using the Or42b receptor mutant: why not just state that "*Or42b* mutants respond to ACV" which is precise and accurate, instead of referring in potentially confounding fashion to "silencing of a single ORN class?"

D) Differences between the preferred Wang and Bhandawat conclusions on Or42b function could also be due the different characteristics of the odor stimuli (there are no sharp odor "cliffs" in the 4-field chamber).

2) The 17 parameters are not easily discerned from the text (one has to read a terse description in the Methods section). Consider adding a table early in the main body of the paper, stating clearly what these parameters are, with maybe a supporting visual that captures this parameter.

3) Please include better labels for all of the figures; the objective of these labels should be to save readers the cognitive strain of going backwards and forwards between image and legend. It is very difficult on the reader to do this with complex figures that have a large number of panels, particularly when the different panels and figures seem so similar.

4) Second paragraph: 'Most chemical communication…'; Is this a valid generalization across all individuals at all times?

5) Results, second paragraph: What is attraction index precisely? It does not seem to be defined in Methods. The relevant section of the Methods says "In conventional olfactory behavioral assays such as a T-maze test, results are typically represented as a performance index, which measures the proportion of time that flies spend inside the specifically defined odor zone (reviewed by Davis 2005). We used a parameter similar to the performance index, 'attraction index', as a measure of the relative change in time a fly spends inside the odor zone between before and during the odor stimulation periods." This is inadequate. The Methods then go on to seemingly imply that it is a combination of the 17 parameters, which I believe it is not.

6) In the subsection “Activation of a single ORN class results in modulation of a small subset of motor parameters”, third paragraph: Do ACV0 and ETA8 activate Or42b-ORNs to similar extents?

7) In the subsection “Modulation of behavior inside the odor-zone is strongly affected in the *Ir8a* mutants”, second paragraph: The rate of decay does not seem different between mutant and control in Figure 10. Can you explicitly measure rate of decay?

8) Figure 10: Show tracks of mutant.

---

## [Author Response]

Several revisions are strongly recommended, largely to focus the paper on its key and simple message regarding the complexity and diversity of motor responses to odorants, to clarify the presentation and some details. One major suggestion is to present the interesting data, which are not easily consistent with Semmelhack and Wang's conclusion that Or42b-ORNs mediate ACV attraction. However rather than attempt to strongly argue against their conclusion (readers can make up their own mind based on the data presented), to use these data simply to highlight the specific contribution of the Or42b cells to the complex ACV behavioural responses measured here. A second suggestion is to carefully edit the figures, labels and legends with a view to increasing clarity and ease of understanding.

First, in regards to the contribution of Or42b-ORNs to the behavioral response to the odor: we completely agree with the reviewers that the importance of experiments aimed at understanding the role of Or42b-ORNs is to show that they affect a small subset of motor parameters. We have now made several edits to further emphasize this point.

Second, we have added many more labels to the figures, as well as, added a table to improve the readability of the manuscript.

Specific comments:1) The studies of "single-ORN" induced motor responses are useful in that they demonstrate specific motor responses to very sparse olfactory input. The point that, ACV responses are not the same as those induced by low ethyl acetate concentrations (EA7 and EA8) which activate Or42b is clear and well made. However, the conclusion that Wang and colleagues were wrong to ascribe specific ACV responses to Or42b ORNs is more difficult to make in rigorous fashion.

We agree with the reviewers that we should present the Or42b data in the context of its contribution to the fly’s behavioral response. We have made several edits to the manuscript to focus our Or42b results to this point. Apart from the specific edits resulting from the reviewers’ feedback below, we have added a few sentences in the Discussion to better express the fact that one limitation of this study is that we have not considered the level of activity and the temporal characteristics of the ORN response. We have both the raw behavioral data, as well as, the electrophysiological data to perform such an analysis. But, to perform such an analysis, one needs a hierarchical statistical model which describes the evolution of behavior over time. Nonetheless, this limitation does not affect the primary conclusions of this study: 1) that attraction and repulsion are crude labels for behavioral response to odors and that odors independently modulate multiple motor parameters; 2) different ORN classes only affect a subset of motor parameters.

A) The authors state that "Or42b-ORNs are the only ones activated at these concentrations" and cite previous publications on this point. However, it is always going to be difficult to be 100% certain that all EA7/8 responses are mediated though Or42b. Could previous experiments have lacked the sensitivity to detect small but significant responses in other ORNs? Could EA also activate other unknown sensory cells in other parts of the fly's anatomy (as appears to be the case for Benzaldelyde)? There is no data in this paper to show that flies lacking OR42b receptors do not respond at all to EA7/8 in the current behavioural paradigm.

We have now included data to show that behavioral response to ETA8 is abolished in the *Or42b* mutant (Figure 6—figure supplement 2). ETA7 behavior is also abolished in the *Or42b* mutant (data not shown). We show that the two parameters – angular speed inside and run duration – modulated by ETA8 (EA is referred to as ETA in the manuscript) are abolished in the *Or42b* mutants. This data strengthens the causal relation between Or42b-ORN activation and its behavioral effects.

B) Could there be more detailed information encoded in the finer aspects of the ORN response – its rate of onset, the spike frequency or rate of change of spike frequency – that allow a fly to discriminate between Or42b signals stimulated by EA and ACV?

We have shown that ACV4, ETA8 and ETA7 which activates Or42b-ORNs to different firing rates all produce comparable responses (Figure 6 and Figure 7). So, within the limits of our experimental design the finer aspects of ORN response does not appear to matter in determining behavioral response. The different behavioral responses to ETA and ACV are more simply explained by invoking the fact that different patterns of ORNs are activated in the two cases.

More generally, we agree with the reviewers that spike frequency and temporal characteristics of the response can influence behavioral response. This issue will require further experimentation.

*C) The above two points are partially (and quite strongly) argued against by analysis of the ACV response of flies lacking the Or42b receptor, but again this does not reproduce the experiment by Semmelhack and Wang, who used shits-based silencing of Or42b-ORNs. There are scenarios in which the two different modes of "silencing" can lead to different results. In this regard, "silencing" may not be the correct term for what is accomplished using the* Or42b *receptor mutant: why not just state that "*Or42b *mutants respond to ACV" which is precise and accurate, instead of referring in potentially confounding fashion to "silencing of a single ORN class?"*

We have changed the title of this section. We have also reworded this section to focus our results on the parameters that are affected in the *Or42b* mutant.

D) Differences between the preferred Wang and Bhandawat conclusions on Or42b function could also be due the different characteristics of the odor stimuli (there are no sharp odor "cliffs" in the 4-field chamber).

It is unlikely that sharpness of the odor cliff can explain the differences in behavior. In the assay in which we measure behavioral response to a point source of odor (Figure 10), *Or42b* mutant are strongly attracted to ACV. We have opted to keep this data out because it does not contribute to the main thrust of the manuscript.

But, we agree with the general thrust of the reviewer comments. It is clear that the relationship between ORN activation and behavior appears to be dependent on the assay (there has been discrepancy even before this study). In this sense, this study provides the way forward – if researchers perform a more nuanced analysis of behavior it would be easier to understand how results in one paradigm compare to results in another.

2) The 17 parameters are not easily discerned from the text (one has to read a terse description in the Methods section). Consider adding a table early in the main body of the paper, stating clearly what these parameters are, with maybe a supporting visual that captures this parameter.

We have now added a table that describes the 17 parameters.

3) Please include better labels for all of the figures; the objective of these labels should be to save readers the cognitive strain of going backwards and forwards between image and legend. It is very difficult on the reader to do this with complex figures that have a large number of panels, particularly when the different panels and figures seem so similar.

We have added many more labels to each of the figures. We hope that these labels will make the figures easier to understand.

4) Second paragraph: 'Most chemical communication…'; Is this a valid generalization across all individuals at all times?

We assume that the reviewers are referring to the “unspecialized metabolites in a precise ratio” part of the sentence. We have removed that part of the sentence.

5) Results, second paragraph: What is attraction index precisely? It does not seem to be defined in Methods. The relevant section of the Methods says "In conventional olfactory behavioral assays such as a T-maze test, results are typically represented as a performance index, which measures the proportion of time that flies spend inside the specifically defined odor zone (reviewed by Davis 2005). We used a parameter similar to the performance index, 'attraction index', as a measure of the relative change in time a fly spends inside the odor zone between before and during the odor stimulation periods." This is inadequate. The Methods section then goes on to seemingly imply that it is a combination of the 17 parameters, which I believe it is not.

Apologies for the confusion; we have clarified this issue. We are now defining attraction index clearly in Table 1 the Methods. In the second paragraph of the Results, we are just referring to fractional time spent inside. Attraction index is now just part of the 17 parameter description.

6) In the subsection “Activation of a single ORN class results in modulation of a small subset of motor parameters”, third paragraph: Do ACV0 and ETA8 activate Or42b-ORNs to similar extents?

No, ETA8 activates Or42b-ORNs to much lower firing rates than ACV0. This raises the question whether the differences in behavior could be simply due to different spike rates. This is unlikely to be the case because ETA7 and ACV2 elicit different behavioral responses despite activating Or42b-ORNs to the same level. Besides, mutating Or42b-ORNs eliminates all responses to ETA7/8 but not to ACV0/2. The simplest explanation is that other ORNs activated by ACV are important. We have added a paragraph to discuss this point.

*7) In the subsection “Modulation of behavior inside the odor-zone is strongly affected in the* Ir8a *mutants”, second paragraph: The rate of decay does not seem different between mutant and control in Figure 10. Can you explicitly measure rate of decay?*

This is a very good point. Although the rates of decay are different for the wild-type and the *Ir8a* mutant, the main point is that time inside/transit is higher in the wild-type compared to the mutant flies and remains higher throughout the course of the experiment. We have rewritten the section describing this result.

8) Figure 10: Show tracks of mutant.

We have decided against this because it is quite hard to tell the difference between wild-type and mutant. We also tried encoding time with color. But encoding time with color also did not work well because of much intersection between tracks.